# Are These the Same Apple?
# Comparing Images Based on Object Intrinsics

**Klemen Kotar**\*, **Stephen Tian**\*, **Hong-Xing Yu, Daniel L. K. Yamins, Jiajun Wu**
Stanford University
\*Equal contribution
{klemenk, tians, koven, yamins, jiajunw}@stanford.edu

## Abstract

The human visual system can effortlessly recognize an object under different extrinsic factors such as lighting, object poses, and background, yet current computer vision systems often struggle with these variations. An important step to understanding and improving artificial vision systems is to measure image similarity purely based on intrinsic object properties that define object identity. This problem has been studied in the computer vision literature as re-identification, though mostly restricted to specific object categories such as people and cars. We propose to extend it to general object categories, exploring an image similarity metric based on object intrinsics. To benchmark such measurements, we collect the **C**ommon paired objects **U**nder differen**T E**xtrinsics (CUTE) dataset of $18,000$ images of $180$ objects under different extrinsic factors such as lighting, poses, and imaging conditions. While existing methods such as LPIPS and CLIP scores do not measure object intrinsics well, we find that combining deep features learned from contrastive self-supervised learning with foreground filtering is a simple yet effective approach to approximating the similarity. We conduct an extensive survey of pre-trained features and foreground extraction methods to arrive at a strong baseline that best measures intrinsic object-centric image similarity among current methods. Finally, we demonstrate that our approach can aid in downstream applications such as acting as an analog for human subjects and improving generalizable re-identification. Please see our project website at `https://s-tian.github.io/projects/cute/` for visualizations of the data and demos of our metric.

## 1 Introduction

Human vision is extraordinarily robust against extrinsic factors that affect object appearances. When driving on sunny or snowy days, in familiar streets or new cities, we can recognize and track traffic lights, other cars, and pedestrians effortlessly. Such an ability to recognize an object based purely on its intrinsic properties, irrespective of any extrinsic factors such as lighting and imaging conditions, object pose, and background, is a key aspect of intelligent vision systems.

Existing computer vision systems often struggle with variations in lighting conditions, object poses, and backgrounds [Hendrycks and Dietterich, 2019, Drenkow et al., 2021, Lee and Kim, 2023]. For example, visual perception models for autonomous driving can fail during extreme weather conditions [Michaelis et al., 2019], and current video generation models suffer from temporal consistency of object identities, producing flickering artifacts that are easily noticeable by humans [Ho et al., 2022]. To understand and improve these artificial vision systems, an important step is to measure the similarity between visual contents (such as images) purely based on the intrinsic object properties that correspond to object identities.

37th Conference on Neural Information Processing Systems (NeurIPS 2023) Track on Datasets and Benchmarks.

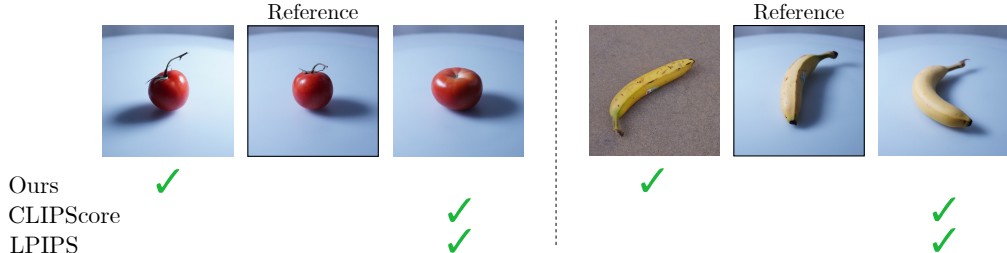

Figure 1: For each reference image, green checkmarks indicate which image a given metric scores as closer to the reference image. Our metric scores images containing the same object as closer than similar but ultimately different ones.

This idea has been studied in the context of re-identification (Re-ID), where the task is to judge whether a pair of images contain an object with the same identity. However, prior approaches to Re-ID are limited to specific categories such as humans [Zheng et al., 2016] or vehicles [Liu et al., 2016]. In this work, we generalize this problem by measuring visual similarities between the intrinsics of *general* objects that define their identities.

While many prior visual similarity metrics have been proposed, we find empirically that they do not serve our goal well, as exemplified in Figure 1. For instance, image-level similarity metrics like PSNR, SSIM [Wang et al., 2004], and LPIPS [Zhang et al., 2018] consider the appearance of the entire image without distinguishing between intrinsic and extrinsic factors. CLIPScore [Radford et al., 2021] focuses on high-level semantic concepts and does not fully account for low-level intrinsic properties of object instances such as shape and texture.

Benchmarking intrinsic object similarity metrics in a systematic way requires images of slightly different objects with the same semantics, combined with varying extrinsics. Existing object-centric datasets that contain similar objects do not capture the same objects with varying poses or lighting conditions. On the other hand, datasets that contain objects with these factors of variation do not include sufficiently similar objects within a single category. This motivates us to collect a new dataset, **C**ommon paired objects **U**nder differen**T E**xtrinsics (CUTE), to evaluate these metrics. CUTE contains $18,000$ images of $180$ objects from semantically similar groups in varying extrinsic conditions such as lighting and orientation as well as in-the-wild images, as shown in Figure 3.

Using the CUTE dataset, we analyze existing similarity metrics and a simple but surprisingly effective framework for approximately measuring intrinsic object similarity. This approach combines foreground filtering and visual features of deep networks trained by self-supervised learning. We conduct an in-depth investigation into which pre-trained features and foreground extraction method provide the best measure of this similarity . From our results, we propose a strong baseline that incorporates two key ingredients that lead to a compact object-centric measurement: using pre-trained representations from DINOv2 [Oquab et al., 2023] and patch-level foreground feature pooling.

Our work represents a step forward in the foundational understanding of how we can model and compute visual similarities based on intrinsic object identity. On one hand, it contributes to the broader scientific objective of creating AI systems that understand the visual world as we do. We show that our measurement approach is able to better approximate the visual similarity perceived by humans than previous metrics [Bonnen et al., 2021]. On the other hand, it holds promise in helping artificial vision systems generalize across different environments and settings. We show that in a visual re-identification task, our metric helps domain-specific vision models generalize better in unseen environments.

In summary, our contributions are as follows: Firstly, we collect a dataset called CUTE containing $18,000$ images of $180$ grouped objects in varying pose, lighting, and in-the-wild settings to benchmark approaches on measuring intrinsic object similarity. We then perform an empirical study of metrics based on pre-trained visual feature extractors to determine which is most effective at measuring this similarity. From this, we propose a simple but strong baseline to measure the similarity called foreground feature averaging (FFA) that combines foreground filtering with features learned by DINOv2 [Oquab et al., 2023], a self-supervised deep model. Lastly, we find that our approach can aid in downstream applications such as acting as an analog for human subjects and improving generalizable re-identification.

## 2 Related Work

**Image similarity metrics.** Many prior methods have been proposed for evaluating the similarity between images, from classical metrics such as peak signal-to-noise ratio to perceptual metrics like the structural similarity index measure (SSIM) [Wang et al., 2004]. Recently, metrics based on deep neural network features have been used to measure image similarity [Salimans et al., 2016, Heusel et al., 2017, Zhang et al., 2018, Sylvain et al., 2021] for training and evaluating generative models. Our metric seeks to compare images based on the intrinsics of objects present in the scene, rather than provide a pixel-by-pixel comparison or image quality assessment.

Concurrent work introduced a text-conditioned metric of image similarity called GeneCIS [Vaze et al., 2023]. It measures a model's ability to adapt to a range of similarity conditions. Our metric on the other hand focuses on a single dimension of similarity - the similarity between the intrinsics of objects. While GeneCIS mines its data from pre-existing datasets and annotates it post-hoc, we collect the data for our benchmark systematically to ensure the ground truth similarity values. DreamSim [Fu et al., 2023] attempts to capture the human notion of visual similarity using a deep neural network trained on a dataset of synthetically generated image triplets with human-annotated similarity preferences. Our metric instead focuses on the ground truth identity of objects, not human preferences.

**Re-identification.** A relevant problem in computer vision is person/vehicle re-identification [Zheng et al., 2016, Liu et al., 2016], which focuses on judging if a pair of person/vehicle images capture the same identity. Early work in this area typically uses hand-crafted visual features [Liao et al., 2015, Liu et al., 2012] with learned classifiers [Prosser et al., 2010] or distance metrics [Zheng et al., 2012], and recent work exploits deep models [Xiao et al., 2016, Sun et al., 2018, Luo et al., 2019, Ming et al., 2022] learned from pair-wise person/vehicle image datasets [Zheng et al., 2015, Wei et al., 2018, Liu et al., 2017, Lou et al., 2019]. In the deep learning era, re-identification has been studied with a supervised learning setup supported by annotated datasets [Li et al., 2014, Zheng et al., 2015, Liu et al., 2017, Bai et al., 2018], and later extended to unsupervised learning [Yu et al., 2017, Fan et al., 2018, Yu et al., 2019, Ge et al., 2020] and a generalizable setup where the testing datasets are not seen during model training [Jin et al., 2020, Zhang et al., 2022]. The intrinsic object-centric image similarity problem that we study can be considered as a generalization of the re-identification problem, where we aim at quantitatively measuring the instance-level similarity between a pair of images of general objects. Our quantitative measurement can be directly applied to the re-identification problem and we evaluate our final metric along with baselines on this problem in Section 5.2.

**Object-centric vision datasets.** Prior efforts have collected visual datasets consisting of different instances of objects from common categories. CO3D (v2) [Reizenstein et al., 2021], Objectron [Ahmadyan et al., 2021], and the Object Scans dataset [Choi et al., 2016] consist of real-world videos of different object instances from typical categories. However, they are not an ideal testbed for our metric as extrinsic factors such as lighting do not vary within a particular video, making it impossible to separate object identity from these cues. The Amazon-Berkeley Objects Dataset [Collins et al., 2022] contains 3D models and catalog images for many items, but these images are not guaranteed to be taken from the real world or even to include the object itself. On the other hand, our dataset contains not only images taken from real-world objects of various categories, but also contains stimuli obtained by varying the lighting conditions and object poses.

**Self-supervised image representation learning methods.** We find that an approach that leverages features learned by deep models trained in a self-supervised manner on large-scale datasets is surprisingly effective on this task. While self-supervised methods continue to be developed [He et al., 2019, Chen et al., 2020b, Caron et al., 2020, Chen et al., 2020a, Grill et al., 2020], we find that DINOv2 [Oquab et al., 2023] yields the best results on our task. While Amir et al. [2021] have explored the use of DINOv1 [Caron et al., 2021] features as dense visual descriptors for co-segmentation and correspondences, in this work we explore how these features can be used to compute intrinsic object-centric similarities.

## 3 Measuring Intrinsic Object-Centric Distances

**Practical measurement.** We aim at a metric such that for a given image of an object, no other object should have a lower distance than any image of the same object. To this end, we have conducted a thorough evaluation of approaches based on deep features learned from self-supervised training,

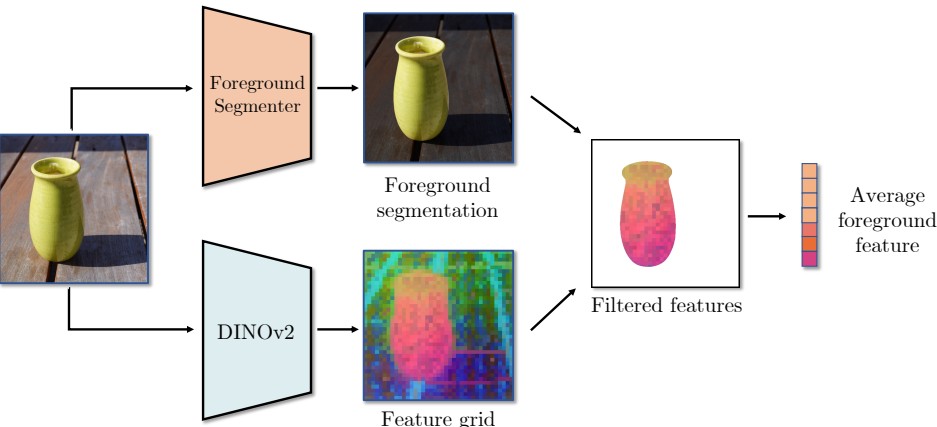

Figure 2: The Foreground Feature Averaging (FFA) (Crop-Feat) pipeline. Each input image is resized to a 336px square, then passed through the DINOv2 [Oquab et al., 2023] model as well as a foreground segmentation model. The foreground features are averaged and compared to another image via cosine similarity. The segmentation is computed at a higher resolution for this visualization.

which we present in Section 4.2. Informed by this, we propose a simple yet effective approach based on deep features learned from self-supervised training, called foreground feature averaging (FFA), as a strong baseline for measuring intrinsic object similarity. Our approach computes embeddings for each image using pre-trained deep models and then computes a similarity score from those embeddings. We show an illustration of our approach in Figure 2.

**Generalizable pre-trained deep feature backbone.** What types of embeddings will enable our metric to exhibit invariances to extrinsics, and generalize to arbitrary objects? Our preliminary exploration showed that features obtained from text-aligned encoders trained on internet scale data such as CLIP [Radford et al., 2021], which have become common backbones in many modern vision pipelines, oftentimes perform poorly at measuring the similarity between highly similar objects. We hypothesize this is because it can be difficult to describe the difference between two similar objects semantically, even when humans can differentiate between them based on visual features.

On the other hand, we find that contrastively-trained models such as DINOv2 [Oquab et al., 2023] produce features that contain a large amount of instance-level information, which is useful for measuring intrinsic object similarity , so we utilize it as the backbone of our metric. DINOv2 encodes input images into patch-level features as well as a global feature. We experiment using both the global feature and the patch-level features. We do not fine-tune the DINOv2 model – our new dataset is used purely as a test set to avoid any possible overfitting brought by fine-tuning.

**Foreground filtering.** Another challenge in computing object-centric similarity is the influence of the image's background, especially when computing image-level features. For our metric, we implement a background cropping procedure that makes our metric more robust for small objects and images with complex backgrounds (such as grass or benches). We analyze two practical instantiations of foreground filtering: (1) cropping the image with off-the-shelf segmentation models before computing features (Crop-Img) and (2) computing deep features on the entire image and then discarding the patch features associated with the background (Crop-Feat). We perform foreground cropping with an off-the-shelf foreground segmentation model, CarveKit [Selin, 2022].

Finally, we compute the cosine similarity between these embeddings for each image as a inverted proxy for the intrinsic object-centric image similarity. This enables non-parametric classification by simply considering image pairs with the highest cosine similarity as representing images of the same object. We implement our metric in PyTorch [Paszke et al., 2019] and release an easy-to-use implementation of it at `https://github.com/s-tian/CUTE`.

## 4   Dataset and Benchmarking

### 4.1   The CUTE Dataset

In this section, we describe the composition and capture of our new dataset, CUTE, that allows for systematic evaluation of intrinsic object-centric similarity metrics. The dataset can be downloaded at `https://purl.stanford.edu/gj714cj0414`.

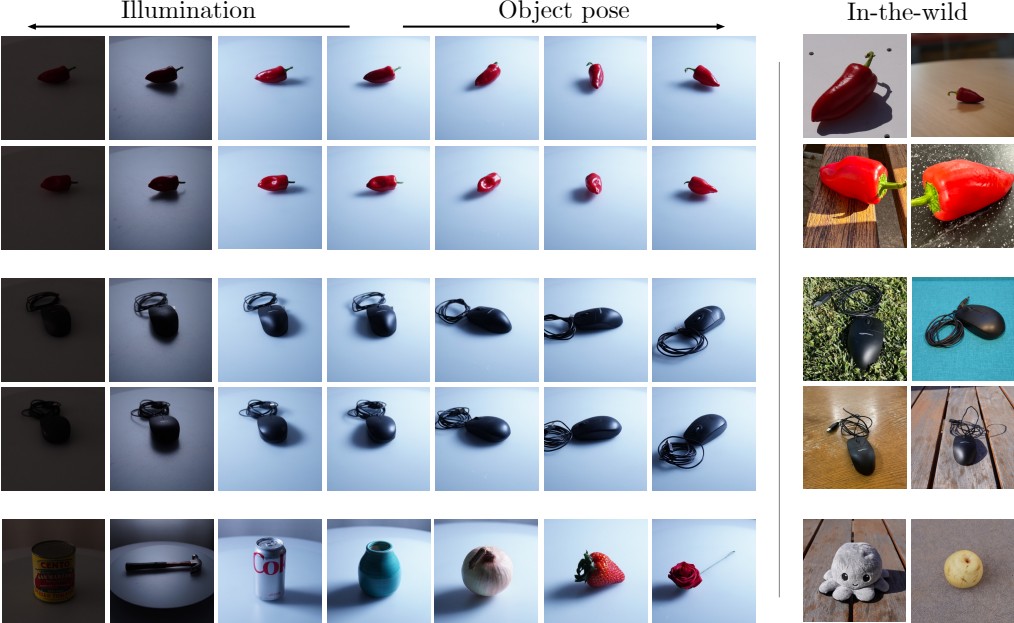

Figure 3: Example images from the CUTE dataset. The top four rows show images captured for paired objects in different illumination conditions and object poses. The last row shows random samples of object instances from across the dataset.

**Category configuration.** Our dataset is composed of *object instances* from each of 50 semantic object categories. Objects belonging to a particular semantic category have differences such as shape or texture that identify them as distinct objects. For example, the category "apples" contains apples with distinct shapes and textures. The dataset contains a total of 180 objects. These categories range from fruits like "apples" and "oranges", to household items like "plates" and "forks". Examples of the objects in the dataset are shown in Figure 3, and a full list of object categories can be found in the appendix. Of the 50 object categories, 10 categories contain ten object instances for additional intra-class variation, while all other categories contain two object instances.

**Extrinsic variations.** To test the ability of metrics to isolate object intrinsics from visual stimuli, which are composed of both intrinsics and extrinsics, we collect images of each object with varying extrinsic parameters. In particular, for each of the 180 objects, we consider three types of extrinsic configurations: different illumination, different object poses, and in-the-wild captures. For different illumination and object poses, we consider a studio setup, where we have 4 lighting conditions and 24 object poses, leading to a total of $4 \times 24 = 96$ images of the same object. We capture in-the-wild images under 4 different environments using different cameras (i.e., different imaging systems). Therefore, our CUTE dataset consists of a total of 18,000 images. In the following, we provide details on our capture setup. More imaging details are in Appendix A.

- **Different illumination.** For the studio capture configurations, we place each object at the center of a turntable in a controlled environment. We capture images using a Sony $\alpha$ 7IV mirrorless camera equipped with a FE 24-70mm F2.8 GM lens. We vary the illumination of the object by turning on each of three LED light panels placed to the left, right, and back of the stage in turn, or turning all of them off, for a total of four lighting settings.

- **Different object poses.** We vary the orientation of each object by rotating it on an automatic turntable at a constant velocity and capturing an image approximately every 15 degrees of rotation.

- **In-the-wild capture.** Finally, for each object, we collect "in-the-wild" images in both indoor and outdoor settings. We collect four images for each object, each in a different location, and with varying imaging conditions: a combination of mobile phones and the aforementioned mirrorless camera. This represents a more realistic distribution of images.

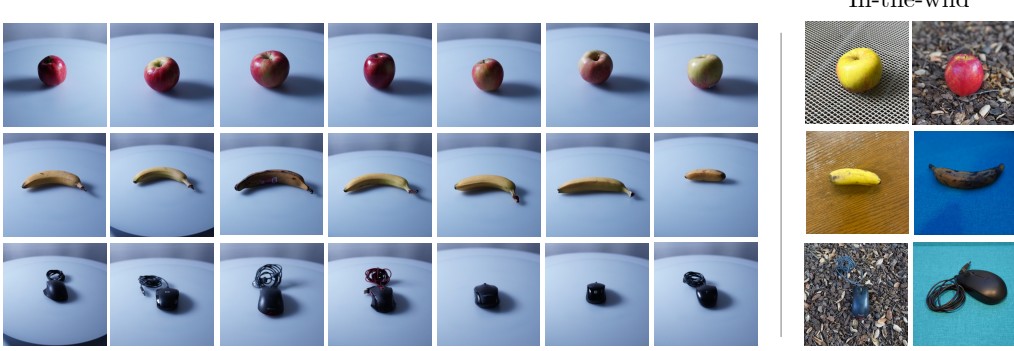

Figure 4: Example images from the *apple*, *banana*, and *mouse* categories. Our dataset contains 10 object instances each for 10 specific categories.

## 4.2 Evaluation

In this section, we profile the performance of popular image similarity metrics and similarities computed using several deep self-supervised learning backbones, including our proposed FFA metric, using the CUTE dataset. We use a subset of CUTE with two object instances per category to ensure a balanced evaluation.

We perform a group-based evaluation, measured by image retrieval (quantified by top-1 accuracy and mean average precision (mAP)). The retrieval candidate with the maximum score is predicted as containing the same object. For CLIPScore and ours which are embedding-based, we also use K-means (with $K = 2$) to cluster embeddings to measure how well the embedding space gets separated according to object identity (quantified by the adjusted Rand index [Rand, 1971] (ARI)). We use separate groups to evaluate the sensitivity of metrics for each of the following scenarios:

- **Illumination**: Each group contains 8 images, 4 containing one object in 4 different lighting conditions, and 4 containing the paired object in 4 different lighting conditions. The pair of objects have similar poses. Thus, 24 groups are formed for each object pair, one for each object pose. In each group, we iterate over every image as the query, and the other 7 as retrieval candidates.

- **Pose**: Each group contains 48 images and is formed by 24 images of one object from each of 24 poses, and 24 of the paired object from 24 poses. Four groups are formed per object pair, corresponding to each lighting condition. In each group, we iterate over every image as the query, and the other 47 as retrieval candidates.

- **In-the-wild**: Each group contains 8 images, with 4 images from different scenes for each of the paired objects. So we have 1 group for each pair. In each group, we iterate over every image as the query, and the other 7 as retrieval candidates.

- **All**: Each group contains all 200 images for a given object pair. We iterate over every image as the query and use all other 199 images as retrieval candidates.

**Results.** As Table 1 shows, our FFA approach compares favorably to prior image similarity methods at most of the evaluation groups. In particular, in the in-the-wild group which is particularly difficult, SSIM and LPIPS are no better than random guess (their top-1 accuracy is around $50\%$), and CLIPScore suffers from different object in the same environment. For controlled groups (illumination and pose), although the top-1 accuracies are above 60% and 90% respectively for both FFA and CLIPScore, the retrieval results are far from optimal reflected by imperfect mAP. The clustering results show low ARI scores for both CLIPScore and ours, suggesting that both cannot separate the embedding space very well, leaving clear room for improvement. Nevertheless, our simple approach performs better at most evaluation tasks.

In Figure 5, we show confusion matrices for a few representative examples from each of the first three groups. In these examples, we show that our metric is able to score images that contain the same object more highly than images containing subtly different objects, even when conditions such as lighting and object pose vary. In contrast, LPIPS is highly sensitive to lighting and background, as

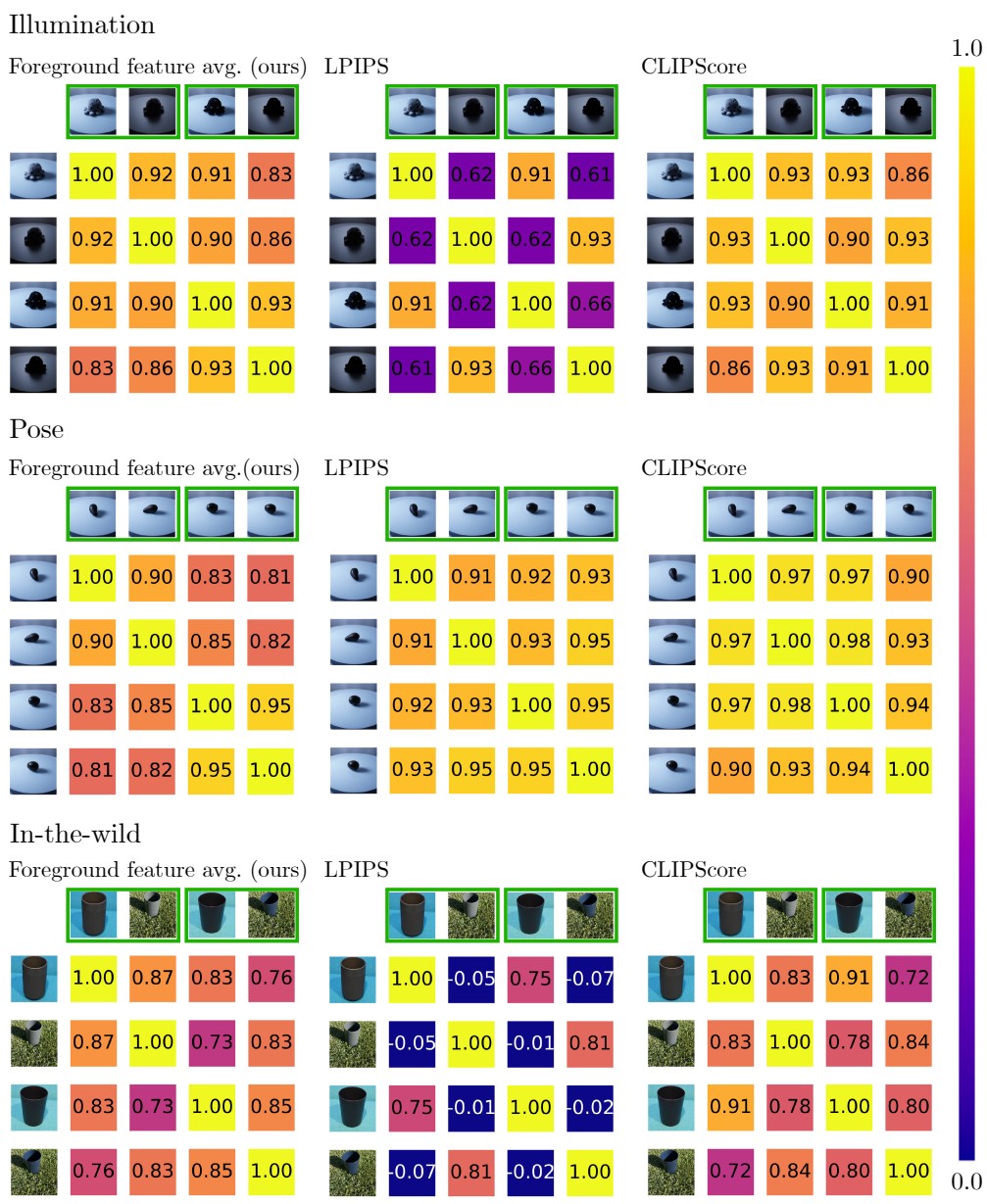

Figure 5: Qualitative visualizations of evaluations on our dataset. Each grid depicts the pairwise *similarity* between sets of four images from each group. Images in green frames contain the same objects. An ideal metric would score all values in the upper left and lower right quadrants of each grid, which are comparisons between the same objects, most highly (closer to 1). Our metric displays this behavior, while LPIPS and CLIPScore do not. Please zoom in to see the visual differences between each pair of objects. The displayed images are cropped for better visibility in this visualization only.

| Setting | In-the-wild | | | Illumination | | | Pose | | | All | | |
|---|---|---|---|---|---|---|---|---|---|---|---|---|
| | mAP↑ | top-1↑ | ARI↑ | mAP↑ | top-1↑ | ARI↑ | mAP↑ | top-1↑ | ARI↑ | mAP↑ | top-1↑ | ARI↑ |
| LPIPS | 38.7 | 49.1 | - | 41.6 | 53.9 | - | 38.2 | 49.3 | - | 39.6 | 49.8 | - |
| SSIM | 40.1 | 49.5 | - | 39.4 | 52.1 | - | 40.3 | 50.6 | - | 39.9 | 50.0 | - |
| CLIPScore | 54.4 | 13.3 | -8.7 | 62.0 | 38.8 | -1.0 | 82.8 | **98.2** | 48.6 | 65.2 | 94.9 | -0.1 |
| CLIPScore + Crop | 69.3 | 59.5 | 15.4 | 69.3 | 75.0 | 11.5 | 77.7 | 94.7 | 38.7 | 68.1 | 94.5 | 18.6 |
| DreamSim | 56.9 | 11.8 | -6.3 | 60.7 | 38.5 | -11.0 | 82.7 | 96.5 | 48.2 | 64.3 | 93.2 | 3.6 |
| DreamSim + Crop | 66.8 | 47.0 | 6.0 | 69.4 | 72.0 | -4.0 | 80.6 | 96.4 | 39.9 | 66.8 | 95.0 | 3.7 |
| DinoV2 | 70.9 | 47.3 | 19.8 | 86.0 | 82.3 | 54.0 | **85.2** | 97.2 | **56.1** | **78.4** | 95.7 | 40.3 |
| DinoV2 + Crop | **79.1** | **69.0** | **41.0** | 84.0 | 85.5 | 39.8 | 83.1 | 96.4 | 53.0 | 76.2 | 96.1 | 36.6 |
| DinoV1 | 52.6 | 3.8 | -8.9 | 58.6 | 37.0 | -10.8 | 81.8 | 97.2 | 47.2 | 63.6 | 93.5 | 1.8 |
| DinoV1 + Crop | 64.4 | 37.8 | 7.1 | 67.0 | 63.0 | -4.7 | 79.6 | 96.5 | 38.0 | 66.4 | 94.8 | 4.0 |
| Unicom | 63.4 | 37.0 | 4.8 | 64.9 | 46.5 | 6.7 | 81.9 | 97.1 | 48.5 | 66.6 | 94.8 | 9.7 |
| Unicom + Crop | 73.9 | 66.3 | 25.5 | 72.2 | 72.3 | 11.3 | 78.0 | 93.8 | 39.5 | 67.9 | 93.5 | 15.1 |
| FFA Crop-Img (ours) | 76.7 | 68.3 | 35.5 | 81.7 | 83.0 | 48.2 | 79.1 | 95.2 | 41.5 | 72.2 | 94.9 | 26.8 |
| FFA Crop-Feat (ours) | 75.1 | 61.8 | 27.2 | **88.8** | **89.0** | **62.9** | 81.9 | 96.6 | 46.8 | 77.1 | **96.2** | **40.5** |

Table 1: Performance of image similarity metrics on differentiating object instances from the paired object section of the dataset. We evaluate them by image retrieval, measured by top-1 accuracy (%) and mean average precision (mAP, %). For embedding-based approach including ours and CLIPScore, we also evaluate it by clustering measured by the adjusted Rand index (ARI) which is multiplied by 100 in the table with 100 indicating perfect clustering, 0 indicating a random clustering and -50 indicating worst-case clustering.

can be seen in the "illumination" and "in-the-wild" groups. CLIPScore scores the highest at matching objects to different poses (though not by a wide margin) while performing worse than our metric - often significantly - in all other categories.

We also evaluate the DreamSim [Fu et al., 2023] metric on our dataset and find that it performs similarly to CLIP. To ablate the effect of foreground cropping, we also test both CLIPScore and DreamSim in a pipeline with the CarveKit [Selin, 2022] as in our FFA pipeline. We find that adding cropping decreases the performance of CLIPScore and DreamSim in the pose category and increases the performance in the other categories. This suggests that cropping the background tends to decrease performance in the pose category - perhaps due to the high similarity of the images and imperfect background removal.

## 5 Applications

### 5.1 Human Subject Analog

We also evaluate foreground feature averaging (FFA) on a dataset of stimuli used in prior studies of human perception mechanisms [Bonnen et al., 2021]. In the study, human subjects are presented with a panel of four images of similar objects and asked to pick the odd one out. The objects are divided along two axes as high and low similarity (to other objects in the panel) and as familiar or unfamiliar (familiar objects are similar to those seen in everyday life, while unfamiliar objects are more abstract). Figure 6 contains a sample of these images. This task was designed to study the effects of the Medial Temporal Lobe (MTL) in image processing, so results were collected for both humans with normal MTL functions and humans with lesioned MTLs.

For neural networks, we encode each image and measure which embedding lies farthest from the others and select it as the odd image out.

**Findings.** We report the results in Table 2. We find that our model outperforms the previously studied best baseline (ResNet-50) [Bonnen et al., 2021], and approaches the performance of human subjects with a lesioned MTL, who rely only on the ventral visual stream (VVS). This forces them to operate closer to a single forward pass in a neural network. Human subjects with a limited 200ms exposure to the stimulus exhibit similar behavior, in line with the findings that the low similarity object discrimination task requires only the VVS. We also evaluate the performance of directly using the DINOv2 class token, and find that it outperforms forward feature pooling in two of the subsets.

These findings suggest that our model approaches human-level performance on tasks that require only the VVS, but falls far short on tasks that require the MTL. We also demonstrate that a large gap

| Similarity | Familiar | | Unfamiliar | |
|---|---|---|---|---|
| | High | Low | High | Low |
| ResNet-50 | 0.28 | 0.90 | 0.18 | 0.72 |
| FFA Crop-Feat (ours) | 0.40 | **0.93** | 0.27 | **0.80** |
| DINOv2 | **0.42** | 0.91 | **0.32** | 0.74 |
| Human (lesioned MTL) | 0.50 | 0.95 | 0.20 | 0.95 |
| Human | 0.90 | 0.95 | 0.80 | 0.98 |

Table 2: Classification accuracy of deep neural networks (ResNet-50 is used as the best baseline from Bonnen et al. [2021]) and human subjects on oddity visual discrimination tasks.

| Metric | VeRi | | CityFlow | |
|---|---|---|---|---|
| | top-1↑ | top-5↑ | top-1↑ | top-5↑ |
| SpCL | 43.0 | 57.8 | 26.7 | 35.2 |
| DINOv2 + Crop | 44.6 | 59.5 | **33.5** | **40.1** |
| FFA (Crop-Img) | 41.0 | 55.7 | 28.6 | 36.1 |
| SpCL+LPIPS | 41.6 | 56.5 | 24.0 | 33.1 |
| SpCL+DINOv2+Crop | **46.1** | **60.5** | 32.6 | 39.6 |
| SpCL+FFA (Crop-Img) | 44.0 | 58.2 | 27.7 | 35.8 |

Table 3: Generalizable vehicle Re-ID results using different distance metrics including a strong Re-ID model SpCL [Ge et al., 2020].

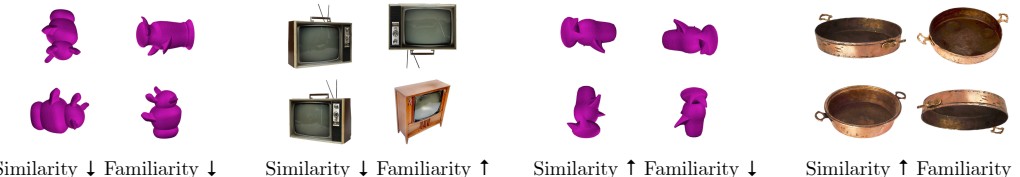

Similarity ↓ Familiarity ↓     Similarity ↓ Familiarity ↑     Similarity ↑ Familiarity ↓     Similarity ↑ Familiarity ↑

Figure 6: Examples of stimuli presented to humans and our metric. Tasks vary across two dimensions: similarity and familiarity. In each task, the human or model is presented with four objects and asked to determine which object is different from the others.

in difficulty exists between the high and low similarity discrimination tasks, even with cutting-edge neural networks trained on hundreds of millions of images, just as they do with humans. Additionally, our approach outperforms humans with lesioned MTL on the Unfamiliar High similarity test case, suggesting that the model is less sensitive to familiarity of objects than human subjects. Overall, our approach is a step towards improved artificial analogs of human subjects, encouraging further studies to eventually create AI systems that understand the visual world as humans do.

## 5.2 Vehicle Re-Identification

In this section, we use our similarity measurement based on FFA to augment the performance of existing models on a generalizable re-identification (Re-ID) task. Specifically, we show how FFA can be used to achieve stronger *transfer* performance on a vehicle Re-ID task. We investigate transfer scenarios between two datasets: VeRi [Liu et al., 2016] and CityFlow-ReID [Naphade et al., 2021]. This transfer setup is referred to as generalizable Re-ID [Jin et al., 2020]. For the CityFlow dataset, as test labels are not provided, we construct a validation set from 100 randomly selected vehicle IDs and randomly sample 1000 query images from that subset.

**Results.** We report the results in Table 3. While the two datasets both contain images of vehicles tracked across multiple cameras, many highly-performant models like self-paced contrastive learning (SpCL) [Ge et al., 2020] do not generalize very well to different datasets due to domain gaps such as different lighting conditions, camera view angles, and background clutter [Song et al., 2020]. To improve the generalization of the model, we augment SpCL with our FFA model measurement via a simple late fusion. Specifically, we compute an $\alpha$-weighted average of the distances computed by the SpCL model and cosine similarity distances computed by our metric. We set the $\alpha$ value to 0.6 for VeRi and 0.9 for CityFlow after tuning over $\alpha = [0.1, 0.2, ..., 0.9]$ on a validation set. As shown in Table 3 and qualitatively in Figure 7, our distance metric aids SpCL in generalization on the VeRi dataset and directly outperforms it on CityFlow.

## 6 Conclusion

In this work, we study the problem of measuring object image similarity based solely on the intrinsic properties of the object. We find that a simple approach based on deep features from self-supervised learning can effectively capture the object intrinsics without being distracted much by extrinsic

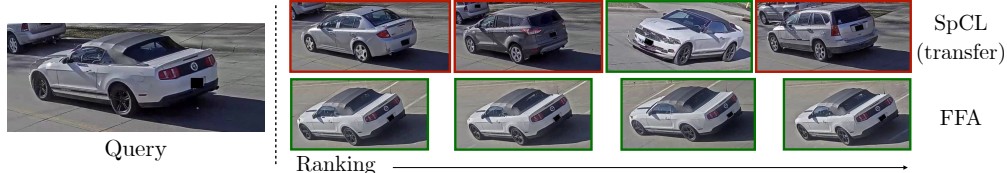

Figure 7: Using our foreground feature averaging (FFA) model directly yields improved vehicle re-identification performance compared to a SpCL model transferred to this task on the challenging CityFlow dataset. In this example, FFA retrieves the correct vehicle in the top four candidates while SpCL only re-identifies it correctly at the third best prediction.

variations such as lighting, object pose, and backgrounds, to approximately measure the similarity. Evaluation on our CUTE dataset validates this finding, yet it also shows that there is room for future improvements. We demonstrate that our approach, with the goal to measure such intrinsic object-centric image similarity, can better approximate human visual similarity understanding behavior, showing the promise to contribute to creating artificial agents that perceive the visual worlds as we do. We hope our findings encourage further explorations in exploiting such similarity for other applications, such as consistent video generation.

**Limitations.** Our approach is focused and evaluated on images containing single objects. The CUTE dataset does not contain images depicting occluded objects or with object poses varied relative to the turntable. Their capture conditions are limited in terms of geographic and outdoor lighting conditions, and while we attempt to vary the capture devices themselves, only four total devices are used.

## Acknowledgments and Disclosure of Funding

We thank Kyle Sargent for helpful discussions. This work is in part supported by Air Force Office of Scientific Research (AFOSR) YIP FA9550-23-1-0127, ONR MURI N00014-22-1-2740, NSF CCRI #2120095, and Amazon, Bosch, Ford, and Google. ST is supported by NSF GRFP Grant No. DGE-1656518.

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

# A    Dataset Details

Our dataset consists of a total of 50 categories with 2 or 10 instances each, for a total of 180 objects. The specific categories are listed below:

1. Apple
2. Strawberry
3. Orange
4. Pear
5. Apricot
6. Banana
7. Mango
8. Broccoli
9. Carrot
10. Potato
11. Yellow onion
12. Shallot
13. Tomato
14. Grapes
15. Lettuce
16. Avocado
17. Bell pepper (yellow)
18. Bell pepper (orange)
19. Bell pepper (red)
20. Can of tomatoes
21. Eggplant
22. Green water bottle
23. Marker
24. Pencil (blue)
25. Pen
26. Mug
27. Ceramic Mug
28. Fork
29. Spoon
30. Butter Knife
31. Spatula (wood)
32. Cups
33. Ceramic pot
34. Plate
35. Bowl
36. Notebook
37. Book
38. Diet Coke
39. Red soda
40. Rose
41. Cookie
42. Octopus
43. Cardboard box
44. Mouse
45. Keyboard
46. Cable
47. Screwdriver
48. Pliers
49. Hammer
50. Screw

Additionally, ten categories contain ten object instances. These categories are enumerated below:

1. Apple
2. Banana
3. Pen
4. Ceramic Mug
5. Fork
6. Spoon
7. Book
8. Mouse
9. Screw
10. Screwdriver

| Light setting | Shutter speed | F-number | ISO | Focal length |
|---|---|---|---|---|
| Left, back, right | 1/640s | f/2.8 | 2000 | 39mm |
| Low light | 1/20s | f/2.8 | 2000 | 39mm |

Table 4: Camera settings for our dataset.

We capture images using a Sony $\alpha$ 7IV mirrorless camera equipped with a FE 24-70mm F2.8 GM lens. The camera settings used in our **different illumination** and **different object poses** data capture configurations are enumerated in Table 4. We show our capture setup in Figure A1.

For the **different object poses** data capture configuration, we use the turntable to perform a full 360 degree rotation at a constant velocity with a period of approximately 24 seconds. We then configure the camera to capture 24 images shooting at an interval of 1 second.

We capture images in the controlled setting at a resolution of $3168 \times 3168$.

For "in-the-wild" settings, we capture images using cell phones (iPhone 13, iPhone 14 Pro, iPhone 12 Mini) at varying resolutions.

# B    Data

**Dataset description.** We provide a dataset description in a dataset sheet: `https://github.com/s-tian/CUTE/blob/main/datasheet.md`

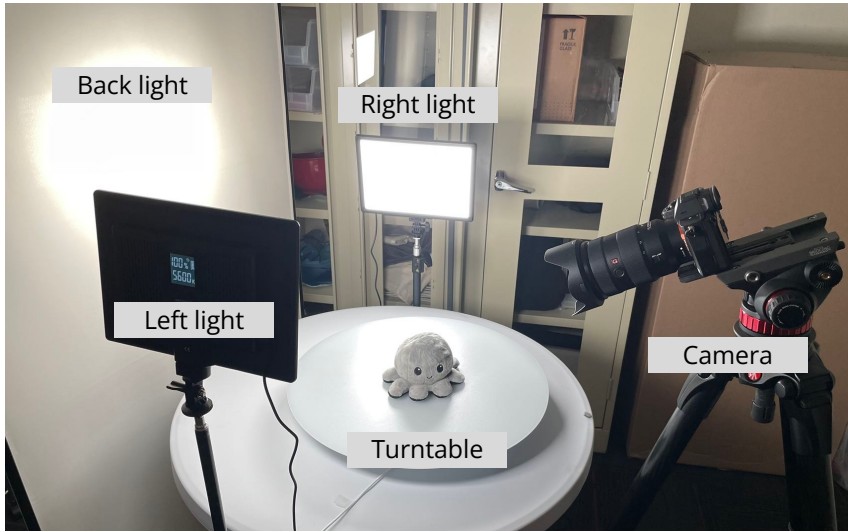

Figure A1: Capture setting for our controlled **different illumination** and **different object poses** configurations.

**Link and license.** The dataset is uploaded for public download under the CC-BY-4.0 license: `https://purl.stanford.edu/gj714cj0414`.

**Maintenance.** Our dataset is hosted on the Stanford Research Data digital repository which will provide long-term support for hosting the dataset. It also provides structured metadata (`schema.org` standards) It has the following DOI: `https://doi.org/10.25740/gj714cj0414`.

**Author statement.** The authors bear all responsibility in case of violation of rights. All dataset images were collected by the authors and we are releasing the dataset under CC-BY-4.0.

**Format.** The data is uploaded in a simple `zip` format. Upon decompressing the archive, a directory is provided for each object category. In each directory, the data is further split into "instance_1" and "instance_2" which represent the two object instances for each category. Under these directories, JPEG images are stored for each lighting condition, with file name descriptors indicating the relative angle in degrees of rotation under which the object was taken, as well as in-the-wild images, which are labeled only with an arbitrary index 0 through 4.

## C  Potential societal impacts

Our paper introduces a metric that may find a range of downstream applications, including improving the temporal consistency of generative models, augmenting vehicle and human re-identification, to aiding perception for embodied agents. As with any improvements, e.g., in consistent video generation, nefarious actors may seek to use these technologies for harmful purposes. We recognize that there may be additional future applications that we cannot currently foresee.

We see opportunities for this research and our findings in Section 5.1 to provide some inspiration for additional research in neuroscience on understanding the human medial temporal lobe.

By introducing a new evaluation dataset, we are also providing a benchmark that may impact the direction of future work. For example, our dataset consists of common objects largely found around North American homes and laboratories, and images are captured in outdoor settings in a limited set of geographical locations. While we select items that we believe are commonly occurring around the world, performing future evaluation on just the CUTE dataset may create a bias towards particular types of objects or scenes. Our metric also largely inherits any biases present in the original DINOv2 model.

# D    Computational resources

The computational resources used were a personal workstation and computing nodes from the Stanford SC computational cluster. We used a personal workstation with an NVIDIA RTX 3090 GPU for the main experiments, and on the SC cluster, we used around 30 jobs lasting at most 2 hours each to perform the Re-ID experiments, including the sweeps on the values of $\alpha$. We use 1 NVIDIA TITAN RTX GPU for each job. We also ran additional backbone ablations on the SC cluster with around 30 jobs lasting at most 4 hours each using one NVIDIA A40 GPU each.

# E    Qualitative Analysis

One characteristic of the proposed metric is its continuous nature. Different from recognition tasks such as Re-ID, it does not simply classify two objects as being the same instance but rather measures how similar they are. Since a measure of distance in the space of all objects is highly subjective and a definitive ground truth is exceedingly difficult to establish, we evaluate our metric using human preference.

Our study design is as follows: one of the authors generated 10 sets of 5 images each, consisting of photos taken by the author and photos from the internet. Then 2 other authors each ranked the image sets, considering both personal preference and the demonstration quality of the set. Their votes were averaged out and the top 5 image sets were selected. These image sets were then scored by LPIPS, CLIPScore, and foreground feature averaging (FFA) and ordered from most similar to least similar to a query object in each set. 34 participants then chose which ordering they prefer according to their personal subjective opinion. The specific prompt they were given was:

"This is a quick, anonymous survey about ordering preference.  Please carefully look at each set of images. 3 Orderings are presented for each set. Select the ordering that makes the most sense to you. The images are ordered from most similar to least similar to the first image (left to right).  There is no one correct answer, we seek your subjective opinion."

In Figure A2 we see that FFA was the top choice on 4 out of the 5 sets and a close second choice on the fifth. While the participants showed strong agreement on certain sets, they generally displayed pretty mixed opinions, highlighting the subjective nature of this type of classification. The given prompt was intentionally vague, allowing the participants to focus on various aspects of the images such as the pose of the objects, the background, the class, etc. Despite this, the results of this limited study suggest that our proposed metric is reasonably aligned with the intuitive human definition of similarity.

# F    Code and instructions for experiments

The code can be found at: `https://github.com/s-tian/CUTE`.

**Experimental details and hyperparameters.** To strike a balance between performance and speed we chose the DINOv2 ViT-B/14 distilled backbone (dinov2_vitb14) for FFA, consisting of 86M parameters. DINOv2 models are capable of accepting inputs at various resolutions, but we select a fixed input size of $336{\times}336$ for the same reason as above, and also to provide a fair comparison to the CLIPScore metric. Our CLIPScore is based on the ViT-L/14@336px model from OpenAI.

In order to obtain the foreground mask we pass the input image through the Tracer-B7 model provided by CarveKit. We then downsample the foreground mask by a factor of 14 (the DINOv2 patch size) in order to obtain a mask of the same size as the DINOv2 feature grid. We then superimpose the two and average the unmasked patches.

# G    Re-ID Experimental Details

We select the hyperparameter $\alpha$ for the weighted sum between SpCL and each considered model by sweeping over the values $[0.1, 0.2, ..., 0.9]$ on a validation set and picking the $\alpha$ value with the best top-1 accuracy on the validation set. The score is always computed by summing $\alpha$ times the model

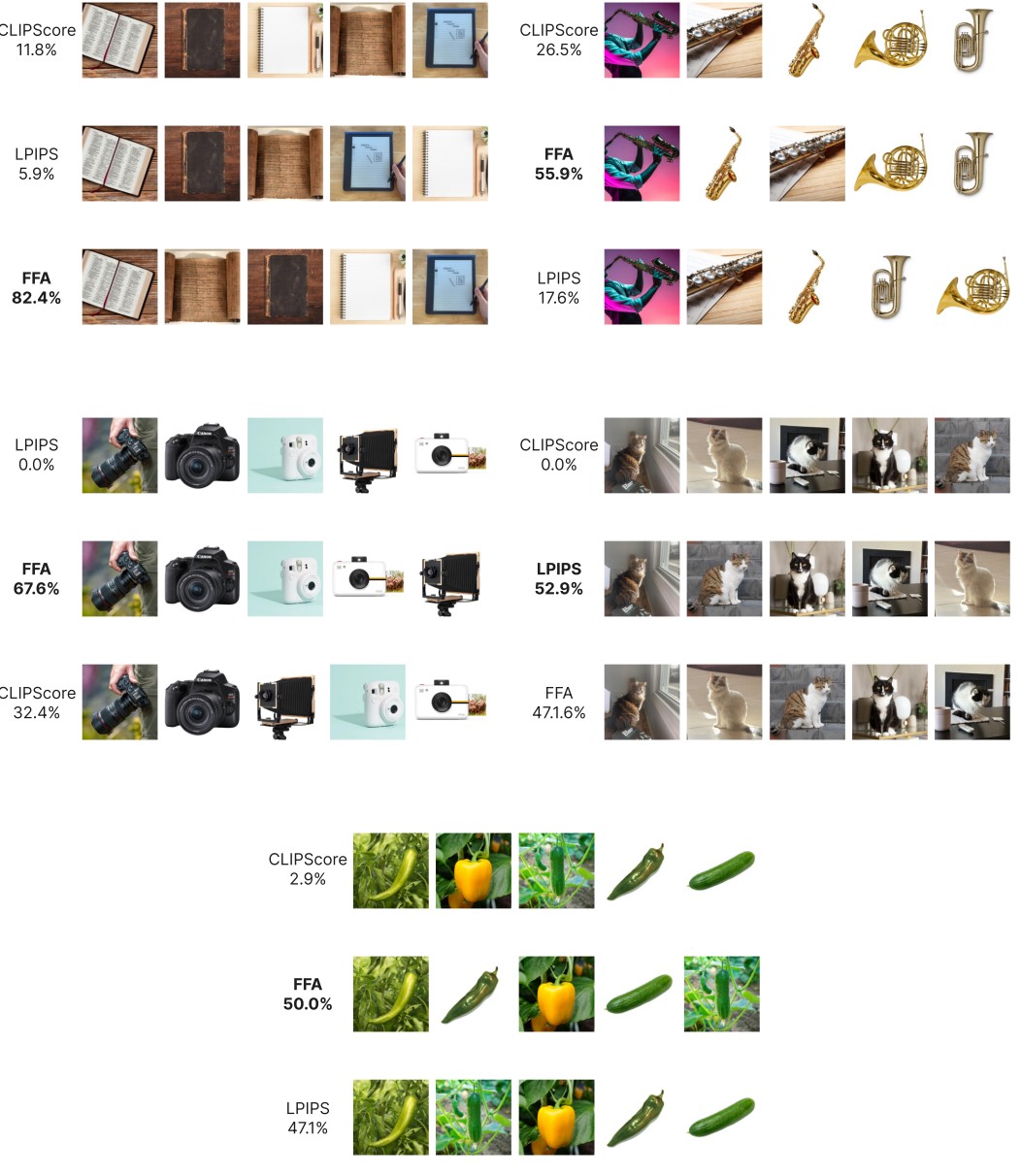

Figure A2: Human survey results. Each of the five groups of images represents an image set. For each image set, we generated three orderings, based on LPIPS, CLIPScore, and foreground feature averaging (FFA) respectively. For each image set, participants were asked to determine which of the three orderings they preferred, where the ordering represented a ranking of the similarity of the first image to the others. We see that in four out of five cases, participants preferred orderings scored as in FFA.

in question and $1 - \alpha$ times the SpCL score. The selected values of $\alpha$ for each model are reported below:

| Model | VeRi | CityScapes |
|---|---|---|
| SpCL+LPIPS | 0.1 | 0.1 |
| SpCL+DINOv2+Crop | 0.5 | 0.9 |
| SpCL+DINOv2 (Global) | 0.1 | 0.2 |
| SpCL+FFA DINOv1 (Crop-Img) | 0.7 | 0.9 |
| SpCL+FFA (Crop-Img) | 0.6 | 0.9 |

Table 5: Selected $\alpha$ values for ReID experiments combining SpCL with various metrics.

We also conducted comparisons to additional models and ablations which were not included in the main text due to space. We report the full results in Table 6.

| Metric | VeRi | | CityFlow | |
|---|---|---|---|---|
| | top-1↑ | top-5↑ | top-1↑ | top-5↑ |
| SpCL | 43.0 | 57.8 | 26.7 | 35.2 |
| LPIPS | 22.5 | 36.1 | 3.5 | 8.0 |
| SSIM | 6.1 | 14.0 | 4.2 | 9.1 |
| DINOv2 (Global) | 16.0 | 26.5 | 15.3 | 23.0 |
| DINOv2 + Crop | 44.6 | 59.5 | **33.5** | **40.1** |
| FFA (Crop-Img) | 41.0 | 55.7 | 28.6 | 36.1 |
| SpCL+LPIPS | 41.6 | 56.5 | 24.0 | 33.1 |
| SpCL+DINOv2+Crop | **46.1** | **60.5** | 32.6 | 39.6 |
| SpCL+DINOv2 (Global) | 42.3 | 56.5 | 24.3 | 33.5 |
| SpCL+FFA DINOv1 (Crop-Img) | 43.6 | 58.1 | 23.9 | 32.7 |
| SpCL+FFA (Crop-Img) | 44.0 | 58.2 | 27.7 | 35.8 |

Table 6: Full version of Table 3, with additional baselines and ablations.

## H  Relation to Other Metrics of Similarity

There are many aspects to object similarity. One can measure the visual similarity - such as the shape, color or texture of objects or functional similarity such as the purpose or affordance of an object. Often these measures are entirely orthogonal to each other, and further influenced by the context of the comparison. Because of this, many measurements of object similarity lack proper ground truth. Our aim is to define one particular dimension of similarity where we can obtain at least partial ground truth labels. We can obtain strong binary labels for this particular metric based on the identity of the objects themselves – different images of the same object should ideally have a perfect similarity.

## I  Evaluations on Extended Dataset

We evaluate all of our pipelines on the extended section of our dataset which contains $10,000$ images - 100 image of 10 instances of objects across 10 categories. Since the result of this 10 way re-identification task is not directly comparable to the 2-way task presented in the paper, we study it here separately. We compute all of the metric as before, except this time we use just the categories containing 10 instances each. As shown in Table 7, the mAP and top1 performance of all of the pipelines is on average higher on this task for deep learning pipelines. We hypothesize this is due to the greater diversity among the 10 object instances (the initial pair of instances was deliberately chosen to be as similar as possible). Furthermore, the ARI scores tend to be much lower because the clustering is performed over ten clusters instead of two. Still, our method obtains the best score on more metrics than any other pipeline.

| Setting | In-the-wild | | | Illumination | | | Pose | | | All | | |
|---|---|---|---|---|---|---|---|---|---|---|---|---|
| | mAP↑ | top-1↑ | ARI↑ | mAP↑ | top-1↑ | ARI↑ | mAP↑ | top-1↑ | ARI↑ | mAP↑ | top-1↑ | ARI↑ |
| LPIPS | 9.0 | 0.0 | - | 11.5 | 0.0 | - | 47.1 | 57.5 | - | 39.9 | 52.0 | - |
| SSIM | 14.8 | 0.1 | - | 11.7 | 0.0 | - | 57.6 | 72.0 | - | 40.3 | 60.2 | - |
| CLIPScore | 84.0 | 82.5 | -0.3 | 85.0 | 85.9 | -0.9 | 87.1 | 97.4 | 8.6 | 84.9 | 96.9 | 3.5 |
| CLIPScore + Crop | 86.0 | 86.6 | 2.8 | 86.5 | 89.9 | 2.8 | 86.2 | 94.3 | 5.43 | 84.9 | 94.3 | 3.3 |
| DreamSim | 84.2 | 81.7 | -0.9 | 84.2 | 85.3 | -2.1 | 87.6 | 96.0 | 6.4 | 84.3 | 95.5 | -0.1 |
| DreamSim + Crop | 86.4 | 86.2 | 5.0 | 86.7 | 90.7 | 0.7 | 86.6 | 95.3 | 5.6 | 84.6 | 95.4 | 2.1 |
| DinoV2 | 85.5 | 84.5 | 2.4 | 88.4 | 91.8 | 5.4 | 87.5 | **97.5** | **9.0** | 85.8 | **97.1** | 6.5 |
| DinoV2 + Crop | **88.3** | **90.3** | **8.2** | 88.0 | 93.0 | 3.9 | 87.6 | 96.4 | 7.0 | 86.4 | 96.5 | 5.3 |
| DinoV1 | 83.1 | 80.4 | -1.0 | 84.2 | 86.6 | -0.1 | 87.6 | 97.1 | 6.1 | 84.3 | 96.8 | 2.9 |
| DinoV1 + Crop | 86.1 | 85.0 | 5.0 | 86.3 | 90.5 | 0.3 | 86.7 | 95.9 | 6.1 | 84.8 | 95.7 | 4.0 |
| Unicom | 85.0 | 84.4 | 2.3 | 84.9 | 86.0 | -0.6 | 87.8 | 96.3 | 7.5 | 66.6 | 94.8 | **9.7** |
| Unicom + Crop | 86.5 | 88.0 | 4.2 | 86.6 | 89.7 | 3.9 | 86.1 | 93.4 | 5.9 | 84.7 | 93.6 | 2.5 |
| FFA Crop-Img (ours) | 87.1 | 88.5 | 3.3 | 88.1 | 92.1 | 5.4 | 86.8 | 95.5 | 7.6 | 85.6 | 95.6 | 6.6 |
| FFA Crop-Feat (ours) | 86.6 | 86.7 | 3.0 | **89.0** | **93.1** | **5.6** | **87.9** | 96.9 | 7.6 | **86.7** | 96.9 | 7.4 |

Table 7: Performance of image similarity metrics on differentiating object instances from our extended 10 instance object dataset. We evaluate them by image retrieval, measured by top-1 accuracy (%) and mean average precision (mAP, %). For embedding-based approach including ours and CLIPScore, we also evaluate it by clustering measured by the adjusted Rand index (ARI) which is multiplied by 100 in the table with 100 indicating perfect clustering, 0 indicating a random clustering and -50 indicating worst-case clustering.

