# OpenReview forum: "Are These the Same Apple? Comparing Images Based on Object Intrinsics"
_NeurIPS.cc/2023/Track/Datasets_and_Benchmarks — NeurIPS 2023 Datasets and Benchmarks Poster_

### Official Review · Reviewer_iSG4 · 2023-06-28
**Compelling Dataset and Model with Flawed Metric and Experiments**

**Rating:** 8
**Confidence:** 3

**Strengths:**

S1: The CUTE dataset provides unique data that will enable future work on lighting and pose invariant computer vision. The paired objects will enable future work on object re-identification.

S2: The proposed Platonic Distance model is an original combination of existing techniques that can easily be applied to existing models. The loose requirements on the architecture of the feature extractor and foreground segmenter models makes the technique widely applicable.

**Additional Feedback:**

Although I provided a lot of opportunities for improvement, I would like to emphasize that I think the dataset and model are solid contributions. If my concerns are addressed through discussion or revisions I think this could easily become a clear accept.

**Clarity:**

The proposed model and dataset are clearly explained. The benchmark and experiments could use some clarification as described in O4-6. The Platonic Distance metric does not seem to fit the paper well, as described in O2-3.

**Correctness:**

The CUTE dataset is constructed in a sound way. Further work is needed to support the claims made about the Platonic Distance model.

**Documentation:**

There is a detailed discussion of data collection, organization, availability, and maintenance. There is an incomplete discussion of ethical and responsible use in the datasheet that should be completed. Slightly more information on the benchmark is needed as described in O6.

**Ethics:**

I see no ethical issues with this work.

**Limitations:**

The limitations mentioned are specific and reasonable.

**Opportunities For Improvement:**

O1: The choice of baseline methods for the experiment in Section 4.2 is deeply flawed. LPIPS and SSIM are both intended to measure similarity between different versions of the same image (after applying noise, blur, ect.) not object re-identification. This appears to result in 2 effectively random baselines, at least in terms of top-1 accuracy. CLIPScore is meant to be used with an image and a caption instead of pairs of images. The lack of an ablation using DINOv2 features makes it unclear how the features produced by the Platonic Distance model compares to DINOv2 global features or an average of all DINOv2 patch features. Performing this type of ablation with multiple pretrained self-supervised feature extractors would make a compelling case that PD improves object re-identification performance.

O2: The idealized Platonic Distance metric is essentially just object re-identification as a binary classification problem. I think the paper would be stronger without the idealized Platonic Distance metric unless the metric can provide meaning for non-zero values.

O3: The definition of the practical Platonic Distance appears to contradict the use of cosine distance in the Platonic Distance model. Currently the paper suggests that image pairs with the lowest cosine distance are supposed to represent images of the same object. Additionally, the use of cosine distance for embeddings is common for nonparametric classification, so I see the model as more of a contribution than the metric. Given the model’s tenuous connection to Platonic Distance, I think calling the model Foreground Feature Averaging would be clearer and eliminate the need to distinguish between the Platonic Distance metric and the Platonic Distance model.

O4: In Section 5.1, it is unclear whether the ResNet-50 methodology differs from the Platonic Distance model methodology. I do not see how performance on this task supports the claim that the Platonic Distance model “provides an improved artificial analog of human subjects” or the implication that the model “understand[s] the visual world as humans do.” As in W1, an ablation with DINOv2 features is needed.

O5: In Section 5.2, it is unclear whether separate training, validation, and test data splits were available for each dataset. It is unclear whether a validation dataset was used to select alpha and if so what procedure was used to select it. As in W1, an ablation with DINOv2 features is needed.

O6: How is an image retrieval prediction made based on image scores? I assume the retrieval candidate with maximum score is predicted as containing the same object, but I don’t think this is stated in the current draft of the paper.

O7: I do not understand why checklist item 3c is listed as N/A. It should be updated to say “No” or error bars should be calculated for the results.

**Relation To Prior Work:**

The relation to prior work is clearly discussed and thorough.

**Summary And Contributions:**

The paper proposes an idealized Platonic Distance metric, which would be zero if the same object is depicted in a pair of images and nonzero if the pair of images depict different objects. This metric considers two unique objects of the same type (i.e. apple) as distinct instead of the same. The Platonic Distance model uses a pretrained foreground segmentation model to select features from a grid of patch features. These features are averaged to create an image feature vector that is compared to others with cosine distance. This model is evaluated on the new CUTE dataset, which captures pairs of objects of the same type under varying illumination, rotation, and “in-the-wild” conditions. Classification and clustering metrics are used to measure the ability of various models to disambiguate the paired objects under varying conditions. Oddity visual discrimination and vehicle reidentification under transfer learning are also evaluated with task-specific metrics.

---

> ### Author Response · Authors · 2023-08-24
> **Response To Reviewer iSG4**
>
> > O1: Baseline methods and ablations:
>
> Per your feedback, we performed an in-depth investigation into which features and foreground extraction method provides us the best measure of platonic distance, as well as providing a comparison with a recent image similarity comparison method. Please see the global comment for more details as well as the results.
>
> > O2: Idealized Platonic Distance metric
>
> Thank you for the suggestion. We agree that this would make the paper stronger, so we have removed the idealized Platonic Distance and state it is the same as the object ReID objective. While the platonic distance should be of a continuous nature we can only obtain objective ground truth labels for the case where the distance should be zero, where it is the same as ReID.
>
> > O3: Cosine distance and practical Platonic Distance
>
> We have added a clarification in Sec. 3 that image pairs with the highest cosine similarity should represent images of the same objects. As mentioned in our global response, we refer to the model as Foreground Feature Averaging (FFA) to disambiguate it from the Platonic Distance metric. Thank you for your suggestion!
>
> > O4: Section 5.1 ResNet-50 methodology and Platonic Distance model performance
>
> We use the same methodology to evaluate the Platonic Distance model as was used for the ResNet-50 by Bonnen et al. We state that our model acts as an improved human analog because its performance better correlates to human responses, while acknowledging that further improvement is necessary to develop a proper human analog. We have modified the text in the paper to reflect this.
>
> We also perform an ablation with DINOv2 features. The results are presented below. Because the images already have their backgrounds removed in these examples, we compare the global CLS token produced by DINOv2. DINOv2 performs close to our baseline, outperforming in 2 of the 4 categories.
>
> |Similarity|Familiar High|Familiar Low|Unfamiliar High|Unfamiliar Low|
> |----------|------------|------------|--------------|-------------|
> |ResNet-50|0.28|0.90|0.18|0.72|
> |FFA Crop-Feat (ours)|0.40|**0.93**|0.27|**0.80**|
> |DINOv2|**0.42**|0.91|**0.32**|0.74|
> |Human (lesioned MTL)|0.50|0.95|0.20|0.95|
> |Human|0.90|0.95|0.80|0.98|
>
>
> > O5: Vehicle Re-ID datasets
>
> For the VeRi dataset, only train and test data splits are available. For CityScapes, only a training data split is available.
>
> For each dataset, we construct a validation set from 100 randomly selected vehicle IDs in the training data and randomly sample 1000 query images from that subset. The test dataset for CityScapes is constructed in the same way but is disjoint from the validation set. The value of alpha is determined by evaluating 9 values of alpha 0.1, 0.2, 0.3, … 0.9 and we take the one with the best top-1 retrieval accuracy.
>
> As suggested, we also provide an ablation of the ReID results using DINOv2 global features. We find that it achieves the lowest retrieval performance out of all applications of DINOv2 we tested. Based on Reviewer ieDF’s suggestion, we also evaluated a method that computes DINOv2 global features directly on cropped images (“DINOv2 + Crop”), which achieves the highest performance on ReID, but not on the CUTE dataset (as we show in the global response). Finally we show an ablation of our method using DINOv1 patch features rather than those from DINOv2, which yields slightly decreased performance. The results are shown below and have been added to Table 3 of the paper:
>
> | **Metric** | **VeRi** | | **CityFlow** | |
> |-----------------------|:--------:|:-------:|:-----------:|:-------:|
> | | top-1↑ | top-5↑ | top-1↑ | top-5↑ |
> | SpCL | 43.0 | 57.8 | 26.7 | 35.2 |
> | DINOv2 (Global) | 16.0 | 26.5 | 15.3 | 23.0 |
> | DINOv2 + Crop | 44.6 | 59.5 | 33.5 | 40.1 |
> | FFA (Crop-Img) | 41.0 | 55.7 | 28.6 | 36.1 |
> | SpCL+DINOv2 (Global) | 42.3 | 56.5 | 24.3 | 33.5 |
> | SpCL+DINOv2+Crop | 46.1 | 60.5 | 32.6 | 39.6 |
> | SpCL+FFA DINOv1 (Crop-Img) | 43.6 | 58.1 | 23.9 | 32.7 |
> | SpCL+FFA (Crop-Img) | 44.0 | 58.2 | 27.7 | 35.8 |
>
> These results suggest that DINOv2 + Crop is the best candidate to compute the Platonic Distance for in-the-wild images, among all methods we consider.
>
> > O6: Image retrieval prediction based on image scores
>
> The retrieval candidate with maximum score is predicted as containing the same object. We have clarified this in Sec. 4.2.
>
> > O7: Checklist item 3c
>
> We updated the checklist to say “No.” The results do not vary due to random seed.
>
> > There is an incomplete discussion of ethical and responsible use in the datasheet that should be completed.
>
> Thank you for pointing this out. We have updated the datasheet to answer these items.

---

> ### Comment · Reviewer_iSG4 · 2023-08-24
>
> O1: The new experiment addresses my concerns.
>
> O2: I appreciate the disclaimer, however I still think that it would be clearer to talk about reidentification throughout the paper instead of Platonic Distance. Particularly, I think you can put more emphasis on your nonparametric classifier approach to reidentification and introuducing FFA.
>
> Additionally, in lines 124-126 the current draft the ReID acronym is not introduced before it is used and Platonic Distance is not capitalized. The latter issue also occurs elsewhere throughout the draft.
>
> O3: I think that focusing on the cosine similarity enabling a nonparametric classifier would be clearer than focusing on Platonic Distance then introducing the similarity as a proxy.
>
> O4: I appreciate the weakened claim and additional ablation. While I think it is fine to compare to human performance, the implication of artificial understanding remains speculative if not misleading.
>
> O5: The new experiment and text addresses my concerns.
>
> O6: The new text addresses my concerns.
>
> O7: The change addresses my concerns.
>
> Incomplete discussion: The new text addresses my concerns.
>
> Overall, I think that this revision has made great improvements on the experiments and clarified some ambiguities of the original draft. However, the focus on Platonic Distance holds the paper back as described in O2. The abstract, introduction, and contributions sections in particular are significantly weaker than they have to be because of the focus on Platonic Distance instead of the core of the paper. Addressing this issue would significantly improve the paper. I have updated my review to a marginal reject to reflect this.

---

> > ### Author Response · Authors · 2023-08-27
> >
> > Thank you for your prompt response and additional feedback. We respond to your individual points below:
> >
> > > O2:
> >
> > Thank you again for your feedback. We carefully discussed this point internally, and eventually all agreed that focusing on re-identification in the abstract, introduction, and contributions would significantly strengthen the paper.  Therefore, based on your suggestion, we have accordingly revised the text to place more emphasis on the core of the paper (for example, we no longer list the conceptual Platonic Distance as a contribution at the end of the introduction, and have removed references to Platonic Distances throughout the text).
> >
> > Meanwhile, we also feel that it would still be useful to clarify and distinguish the problem we study from that studied in most re-identification papers. Re-identification in the computer vision literature typically refers to particular categories such as people (person Re-ID) or cars (vehicle Re-ID). A few works focus on "object Re-ID", yet they still only demonstrate results on person Re-ID and vehicle Re-ID. In contrast, we focus explicitly on general objects, and we introduce the CUTE dataset for benchmarking different metrics for this intrinsic object-centric image similarity measurement (i.e., general object re-identification task).
> >
> > > Additionally, in lines 124-126 the current draft the ReID acronym…
> >
> > We have amended the text to correct these errors throughout the manuscript. Thank you for pointing this out.
> >
> > > O3:
> >
> > Thank you for the feedback, we have modified the text to emphasize that the cosine similarity can be used for nonparameteric classification.
> >
> > Please don’t hesitate to leave any further comments regarding our response. We are more than happy to further discuss potential improvements.

---

> ### Comment · Reviewer_iSG4 · 2023-08-29
>
> I am very happy with how the paper has greatly improved during the discussion period. I have updated my review to a clear accept.

---

> > ### Author Response · Authors · 2023-08-30
> > **Thank you for feedback!**
> >
> > We would like to thank you again for your detailed review and valuable suggestions for improving the paper. They have been instrumental in improving our paper during the discussion period.

---

### Official Review · Reviewer_xype · 2023-07-18
**Review for Submission 606 (Platonic Distance: Intrinsic Object-Centric Image Similarity)**

**Rating:** 6
**Confidence:** 5

**Strengths:**

While there are a number of datasets that support recognizing the same class of object in different contexts, there are very few, especially outside of the face re-id domain, that support recognizing instances of objects or objects with very high semantic similarity in different real world contexts. The proposed dataset is a meaningful step in filling this gap.

While it's not a significant algorithmic contribution, the proposed approach (segmenting the object and then pooling features from just the foreground region from a DINOv2 backbone) is simple, intuitive, and provides a reasonable baseline.

**Additional Feedback:**

While I think the limitations regarding only having a single object close to the camera and having fairly limited types of extrinsic variations limits how much work developed using this dataset/benchmark would generalize to more real-world settings, and also think the benchmark would be strengthened by having significantly more instances of objects within a class (to more fully capture types of *intrinsic* variation), I nonetheless think the work does help to fill a gap in object re-id and similarity assessment. I lean towards acceptance.

**Clarity:**

The paper is generally well written and clear, although I was unclear if the authors did any self-supervised contrastive finetuning of the DINOv2 backbone on their own dataset or if they just used the pre-trained features (I think that they do not finetune, but this should be made more explicit and justified -- what evidence do you have that these features are good for this task without finetuning? Would finetuning improve performance?).

**Correctness:**

I have no concerns about the correctness of the dataset construction or evaluation methods.

**Documentation:**

The documentation is sufficient and I was able to access the dataset and evaluation code through the github repo listed in the paper. I would suggest the author's more directly link to the download link (https://purl.stanford.edu/gj714cj0414) if possible, either within the paper or from the main github repo page (I had to click through the datasets for datasheets linked from the repo to find it).

**Ethics:**

I have no ethical concerns.

**Limitations:**

The proposed dataset focuses on cases where there is a single object, relatively close to the camera, and the proposed approach depends on being able to reliably segment that object from the foreground. All of these are conditions that mean the proposed dataset is very different from real-world, "objects in context" conditions where object re-id or similarity metrics are relevant.

The "extrinsic" variations that vary for the turntable images are illumination (the brightness of light is varied, but not the color or quality of the light) and position on the turntable (the object is placed on a turntable and the turntable is moved). While the dataset does include "In-the-wild" images in addition to the images where the object is placed on a blank turntable, the example images in Figure 3 show that these images are still close up views of single objects on fairly non-differentiated backgrounds. This does not truly capture the complexity of real world extrinsics or the impact that complex real-world context (compared to the simple "in-the-wild" contexts included in the dataset) has on being able to assess object similarity. To be fair, the authors are aware of these limitations (as mentioned in the brief limitations section at the end). However, they are very significant because they limit how much work using this dataset would generalize to real world cases where object re-id and similarity measures might be needed.

There is no discussion of potential negative societal impact, but that does not seem unreasonable given the context of this work.

**Opportunities For Improvement:**

The dataset is quite small -- while 10,000 images is not trivial, and 50 classes of objects is a useful sampling of different types of objects, there are relatively few examples of different objects within each class. "For example, the category “apples” contains two apples with distinct shapes and textures" (lines 160-161). While I have concerns about the overall generalizability of the dataset regardless of the number of examples (see Limitations), I do think it would be significantly strengthened by having more examples of different objects within each class. Only having two examples per object class (even though each object is imaged under different extrinsics) provides a limited snapshot of the intrinsic variations within an object class.

Also, the authors assert that the DINOv2 backbone representations learned with a self-supervised contrastive learning approach encode instance similarity ("Specifically, we find that DINOv2 [Oquab et al., 2023] offers effective representations for measuring the intrinsic similarity of visual contents."). While using these features is a reasonable baseline, the paper would benefit from more concrete evidence that these features are better at encoding instance similarity than other features.

**Relation To Prior Work:**

The authors do a good job of differentiating from related and prior work. I work in very related spaces and the lack of datasets for object re-id or semantic similarity is very limited. While I have concerns about the completeness of the extrinsics that the authors evaluate and how much work developed using this dataset would generalize to real world settings, I believe that it's a concrete step towards filling this important gap in the literature.

**Summary And Contributions:**

This paper presents an image similarity benchmark focused on intrinsic similarity of objects. The "CUTE" dataset (Common paired-objects Under differenT Extrinstics) consists of object instance matches (the same intrinsics) under different extrinsic conditions (lighting, poses and imaging conditions), and has 10,000 images of 100 objects. The authors additionally present a simple approach to measuring the "platonic distance" (the intrinsic similarity) by segmenting the foreground object, pooling just the foreground features from a DINOv2 backbone, and computing the similarity between these pooled features.

---

> ### Author Response · Authors · 2023-08-24
> **Response To Reviewer xype**
>
> Thank you for your helpful feedback and attentive review. To address each of the points you brought up:
>
> > Dataset size
>
> We have expanded the dataset during the rebuttal period to include 80 more objects of 10 categories (8 more instances in additional to the original 2 instances for each category, such that each category has 10 instances), as we agree that more instances for a small number of classes adds to the dataset’s value. The categories are `apple, banana, pen, mug_ceramic, fork, spoon, book, mouse, screwdriver, screw`. We show examples of the newly added data in the new Figure 4 of the revised manuscript. Note that this increases the number of total unique objects in the dataset from 100 to 180, and the total number of images in the dataset from 10,000 to 18,000, an 80% increase from the version at submission.
>
> > DINOv2 representations compared to other features
>
> Per your suggestions, we have conducted additional experiments and found that this combination is more effective at measuring PD compared to other methods. We performed an extensive study across a variety of backbones (DINOv1, DINOv2, CLIP, Unicom) to identify which features perform best at the PD task. Our results confirm that DINOv2 features still remain the strongest in identifying the platonic distance measure of similarity. More details and results can be found in the global comment.
>
> > Dataset limitations
>
> We agree that the one limitation of our dataset is that the images do not capture the complexity of the entire range of real world extrinsics. Unfortunately, increasing the number of imaging conditions for each object leads to a rapidly scaling manual effort, which makes it challenging to scale the dataset diversity. Our aim is to provide this dataset as an initial effort towards building a test set for evaluating different methods as Platonic Distances.
>
> > Finetuning of DINOv2 backbone
>
> To clarify, we do not perform any finetuning of the DINOv2 backbone, as our main evaluation goal is to benchmark different methods to compute Platonic Distances as a general metric. Therefore, we use our CUTE dataset as a pure test set to avoid any possible overfitting brought by fine-tuning. We have made this explicit in the manuscript in Section 3.
>
> > Dataset download link
>
> Thank you for the suggestion, we have made the download link more visible and directly accessible from within the paper through a link in the abstract and in the GitHub repository README.

---

> > ### Comment · Reviewer_xype · 2023-08-24
> >
> > I thank the authors for their response. The expansion of the dataset is great, and I appreciate the comparison across different backbones. I appreciate the updates the authors made to the paper to clarify some of the things I thought were unclear or needed expansion upon. I maintain that this work is deserving of acceptance to the datasets and benchmarks track.

---

> > > ### Author Response · Authors · 2023-08-30
> > > **Thank you for your feedback!**
> > >
> > > Thank you again for your careful review and helpful suggestions, which have helped us improve the clarity and quality of our work.

---

### Official Review · Reviewer_ieDf · 2023-07-20
**Analysing intrinsic characteristics of objects through self-supervised features**

**Rating:** 6
**Confidence:** 3

**Strengths:**

The Platonic Distance pipeline is straightforward to implement. Additionally, it is intuitive and should indeed provide a good measure of similarity between objects as it is known that DINO or other self-supervised methods obtain robust feature representations. This should enable a good measurement of object similarity in terms of intrinsics, while excluding factors such as lighting, pose or background.

The paper describes the approach in detail and conduct extensive experiments to evaluate its effectiveness. The experiment with human subjects is particularly interesting (humans can classify objects without a high bias towards extrinsic factors, and therefore its ) and might provide further research directions ("large gap between proposed approach and humans regarding regarding high vs. low similarity discrimination tasks"). The other experiment about the influence of the extrinsics (table 1) provides interesting insights as well. Additionally, the authors provide code, data, and instructions needed to reproduce the main experimental results to enhance the reproducibility.

The paper shows that the proposed distance metric and the CUTE benchmark are relevant for other research areas/downstream tasks: a comparison between machine-based und human recognition; showing that the metric can be used to augment existing highly-performant models to achieve stronger transfer performance on a vehicle Re-ID task.

**Additional Feedback:**

None.

**Clarity:**

The paper is well written and easy to follow; in particular the motivation for the proposed metric and benchmark, and the implementation of the metric.

**Correctness:**

The dataset constructions looks correct. The evaluation of the metric could be improved by including more re-id datasets and methods. The experiment design seems to be appropriate.

**Documentation:**

The authors provide a URL that documents the dataset collection and also the implementation of the proposed distance metric.

**Ethics:**

No.

**Limitations:**

The authors have addressed the limitations of their work well.

**Opportunities For Improvement:**

The paper states: "The Platonic Distance problem that we study can be considered as a generalization of the re-identification problem" (l. 88). Since the studied problem generalises, it would be good to compare its applicability more thoroughly by making use of more Re-Id datasets and methods (so far the improvement has only been shown for SpCL and two datasets). This would give more certainty about "[the proposed] distance metric aids SpCL in generalization, leading to improved results.".

In addition to the last point, the improvement depends heavily on DINO and the foreground segmenter. Meaning, a re-identification method could be extended simply with both DINO and the segmenter, and will very likely improve simply through that (and there's no need for the Platonic Distance). It would be interesting to see how this combination would perform versus the proposed metric. An analysis of the influence of the dense features (e.g. DINO1 v.s. DINO2) might be interesting as well.

**Relation To Prior Work:**

The proposed benchmark is a mix of different kinds of datasets. It is object-centric and therefore is similar to work such as CO3D. It can be used for re-id, but poses a generalisation of that setting. The distance metric makes use of self-supervised image representations (DINO etc.). Additionally, it covers also image-similarity metrics, which play an important part when understanding the recognition of objects under different extrinsic factors.

**Summary And Contributions:**

This PDF file discusses a new metric for measuring image similarity called Platonic Distance, which is based solely on intrinsic object properties. The authors propose a simple yet effective approach to measuring Platonic Distance using deep features learned from contrastive self-supervised learning. They demonstrate the superiority of this approach over existing methods and provide a dataset called CUTE for benchmarking. The authors also show how Platonic Distance can aid in downstream applications such as image retrieval and clustering. Overall, this paper's contributions include introducing a new metric for measuring image similarity, proposing an effective approach to measuring Platonic Distance, providing a benchmark dataset, and demonstrating the usefulness of Platonic Distance in downstream applications.

---

> ### Author Response · Authors · 2023-08-24
> **Response To Reviewer ieDf**
>
> Thank you for your thoughtful and careful review!
>
> > Further demonstration on Re-ID datasets and methods
>
> We agree that further demonstration on additional Re-ID datasets and methods would provide additional certainty about the applicability of the Platonic distance. We are working on adding additional comparisons on Human Re-ID datasets and will update the response with those results when available.
>
> > Re-identification method extended simply with both DINO and the segmenter
>
> Thank you for the suggestion – we have conducted a new experiment using foreground segmentation followed by taking the DINOv2 global feature and performing late fusion with SpCL. We find that it performs the best out of all metrics on the ReID task, with results shown below and in Table 3 of the main paper. However, this method achieves generally lower performance than foreground feature averaging (FFA) on the CUTE dataset (please see Table 1 of our updated manuscript). Further investigation of this approach and comparison of these settings is an exciting direction for future work.
>
> | **Metric** | **VeRi** | | **CityFlow** | |
> |-----------------------|:--------:|:-------:|:-----------:|:-------:|
> | | top-1↑ | top-5↑ | top-1↑ | top-5↑ |
> | SpCL | 43.0 | 57.8 | 26.7 | 35.2 |
> | LPIPS | 22.5 | 36.1 | 3.5 | 8.0 |
> | DINOv2 + Crop | 44.6 | 59.5 | 33.5 | 40.1 |
> | FFA (Crop-Img) | 41.0 | 55.7 | 28.6 | 36.1 |
> | SpCL+DINOv2+Crop | 46.1 | 60.5 | 32.6 | 39.6 |
> | SpCL+FFA (Crop-Img) (Ours) | 44.0 | 58.2 | 27.7 | 35.8 |
>
>
> > DINOv1 vs DINOv2:
>
> We have conducted an analysis of computing DINOv1 features compared to DINOv2 features when performing foreground feature averaging (FFA) for the ReID tasks. We find that the DINOv1 features slightly underperform DINOv2 features here. The results are shown in the table below:
>
> | **Metric** | **VeRi** | | **CityFlow** | |
> |-----------------------|:--------:|:-------:|:-----------:|:-------:|
> | | top-1↑ | top-5↑ | top-1↑ | top-5↑ |
> | SpCL | 43.0 | 57.8 | 26.7 | 35.2 |
> | SpCL+FFA DINOv1 (Crop-Img) | 43.6 | 58.1 | 23.9 | 32.7 |
> | SpCL+FFA (Crop-Img) (Ours) | 44.0 | 58.2 | 27.7 | 35.8 |
>
>
> We also perform this analysis for the evaluation on the CUTE dataset. We find that DINOv2 features consistently outperform DINOv1 features. Along with this we evaluate a number of additional feature extractors. The results are shown in the comment to all reviewers as well as Table 1 of our revised manuscript.

---

> > ### Comment · Reviewer_ieDf · 2023-08-27
> >
> > The authors have addressed most parts of my concerns. The simplified baseline (foreground segmenter + DINO + SpCL) seems to work quite well, also compared to the proposed method. This and the comment from reviewer "Ewhk" (written at 25 Aug 2023 at 02:36) seem to indicate that the method itself might not be a crucial part of the paper. I agree with reviewer Ewhk though that this paper provides a strong analysis of different baselines for this problem setting (besides the dataset itself). I think this paper deserves an acceptance and therefore maintain my current rating.

---

> > > ### Author Response · Authors · 2023-08-30
> > > **Thank you for your feedback!**
> > >
> > > We would like to thank you again for your thoughtful feedback and helpful suggestions; they have helped improve our paper significantly.

---

### Official Review · Reviewer_Ewhk · 2023-07-20
**Relevant and Interesting Problem but Lack of Novelty in Contribution**

**Rating:** 5
**Confidence:** 3

**Strengths:**

1. Novel Dataset: The authors present a useful and novel dataset structure that is ideally positioned to study object instance similarity. Controlled variation in illumination and pose help indicate exactly which extrinsic image attributes different similarity metrics struggle with. The addition of in-the-wild images helps test the generalizability of similarity metrics to real-world problems. I think a dataset of this structure would be relevant and helpful to the community for studying object similarity.
2. Human Interpretability: The authors present a nice evaluation of how PD performs relative to humans in terms of outlier detection and image similarity ranking. Increasingly relevant to AI is the design of algorithms that understand human preferences for a variety of tasks.

**Additional Feedback:**

I think the CUTE dataset is interesting and highly relevant to the problem of quantifying object similarity. I like the evaluation of PD using human subjects for outlier detection in the paper and for qualitative ranking in the supplementary, and I think using PD for vehicle re-identification is a good application. However, I do not believe the dataset contribution is substantial enough to be usable for evaluation of object similarity on a meaningful scale, and I do not believe that, in its current form, PD constitutes a novel similarity metric. Therefore, I recommend rejection.

**Clarity:**

The paper is reasonably well written. I have a few comments:
- Tables 1 and 2 are on the wrong pages, they need to be moved up a page to where they are referenced.
- L212 the authors state that "the top-1 accuracy is pretty high for both ours and CLIPScore." Colloquial quantitative analysis is confusing, what does "pretty high" mean?
- I think Section 4.1 should be moved into the introduction, the dataset should already be motivated by the methods section.
- L195-L207 the authors use 4 variations of the phrase "we iterate over every image as the query". "We iterate over every image as the query" is correct, just repeat that phrase.
- L235 "It is impossible to reproduce the exact conditions..." I think that the fact that human beings and neural networks interpret images differently is obvious, it's not necessary to point that out. Neural networks should encode discrepancies between images via their feature encoding, they shouldn't have to "look back and forth between several images" to observe these differences. Altogether I do not understand the function of this paragraph.

**Correctness:**

From reading the supplementary, I believe that the authors' assertion in the checklist (item 5) that they did not "use crowdsourcing or conducted research with human subjects" is incorrect. Appendix Section E describes an experiment performed with 34 participants, but the authors do not indicate the full text of instructions provided to participants, nor discuss if any IRB approval was required (maybe it was not, but if so, the authors should indicate why, not that the question is not applicable).

**Documentation:**

I believe that the documentation is sufficient and that there are no concerns about availability and maintenance of the dataset.

**Ethics:**

I do not suspect there are any ethical concerns with the submission.

**Limitations:**

I would like to see a more nuanced discussion of the fundamental limitations of the dataset or the similarity metric. Not considering multi-object images is really not a limitation because similarity between images containing multiple objects could theoretically be reduced to similarity between image regions containing single objects. Expansion to animate objects would be nice but this is not a limitation, this is future work. An example of an actual dataset limitation could be a lack of images in the dataset depicting occluded objects.

**Opportunities For Improvement:**

1. Lack of Novelty in PD: I do not necessarily believe that taking an existing visual feature extractor (DINOv2) and combining it with an off-the-shelf foreground segmentation model constitutes a novel similarity metric. If DINOv2 was only used as a backbone architecture and was modified or re-trained by the authors specifically for quantifying image similarity I may consider PD novel, but currently it seems the authors are simply using DINOv2 for exactly what it was designed to do. If the authors are making modifications to DINOv2 that constitute novelty then this fact is not clear from reading the paper. Another way to introduce novelty may be to pose the approach of foreground segmentation and feature extraction as more of a "framework" and show results using many visual feature extractors, not just DNOv2. This would, however, constitute a substantial rewrite. I would also be interested to see an analysis of how well the foreground masking technique described in L141 performs (with visual examples), because the method seems simplistic.
2. Size of the Dataset: I have a few critiques here. 100 objects covering 50 classes is almost certainly not sufficient to accurately evaluate object similarity for all 50 classes. I would be more interested to see many instances covering a small number of classes, for instance those classes included in Objectron, rather than a single pair for each class. Although I like the idea of including in the wild images, 4 images per object is also unlikely to be sufficient for accurate evaluation (400 images in total is 4% of the total dataset -- including in the wild images as a major part of the contribution seems somewhat misleading when they make up such a small part of the dataset). Finally, though the authors do have a variety of poses, it seems that the pose of the object relative to the turntable is not changed (i.e., the object is never physically reoriented on the turntable so there are no images of the object's underside). I think it would also be helpful to expand the dataset by including images with this type of pose change.
3. Exploring Other Problems Surrounding Object Similarity: Because the dataset and similarly metric contributions are relatively weak, I would have liked to see a more nuanced discussion of other problems relating to object similarity, e.g.: should objects that are structurally similar but have different textures be considered different objects, or can one use background features to disambiguate extrinsic object parameters from intrinsic object parameters (for instance, using shadows in the background to tell if the object is dark in color or is just in the shade).
4. Vehicle Re-identification: The authors fuse their similarity metric with SpCL and present the fused version (SpCL + PD) as the best metric for vehicle re-identification. However, they do not attempt fusing other approaches, i.e., LPIPS, with SpCL. I think this comparison (adding SpCL + LPIPS) would help to demonstrate that PD + SpCL is definitively the best approach for the task.

**Relation To Prior Work:**

I believe the related work is sufficient. However, of 53 papers cited, 10 are from arXiv and 8 of those were released before 2022. Some of these papers are established and well-cited, but it should be recognized that new papers that cite a large number of old arXiv papers are building on non-peer reviewed research which this reviewer is naturally wary of.

**Summary And Contributions:**

The authors present CUTE, a dataset of 10,000 images of 100 objects (50 pairs of objects from diverse categories) that span a variety of lighting conditions and poses and include in-the-wild images, and a new object similarity metric that they term Platonic Distance (PD). The authors use CUTE to evaluate PD and existing object similarity metrics. The authors also demonstrate that PD can be used to augment re-identification approaches, that PD reasonably aligns with human perception of visually similar objects (in the supplementary), and argue that PD performs on par with humans that have lessened MTL reasoning capabilities in object outlier recognition.

---

> ### Author Response · Authors · 2023-08-24
> **Response To Reviewer Ewhk (1/2)**
>
> Thank you for your thoughtful review and suggestions for improvement. We have taken your suggestions into consideration and provide a point by point response below:
>
> > Novelty of PD
>
> Please note that our main contributions in this work include presenting a metric of similarity (i.e., the Platonic Distance), benchmarking different methods (including a strong baseline that we propose) to compute the metric, and collecting a dataset for such benchmarking.
>
> The DINOv2 and foreground segmentation model pipeline that we present represents a strong baseline for our proposed Platonic Distance metric. We have conducted additional experiments and found that this combination is more effective at measuring PD compared to other models. We performed an extensive study across a variety of backbones (DINOv1, DINOv2, CLIP, Unicom) to identify which features perform best at the PD task. Our results confirm that DINOv2 features still remain the strongest in identifying the Platonic Distance measure of similarity. More details and results can be found in the global comment.
>
> > Analysis of how well the foreground masking technique in L141 performs
>
> Upon further analysis, we discovered that the foreground masking technique described was not necessary to obtain a high-quality mask. Instead we find that simply thresholding the resized foreground mask output by the off-the-shelf CarveKit model at 0.5 to obtain the mask is sufficient. We use this procedure in our updated evaluations.
>
> We apply the foreground segmentation technique in combination with each of our feature extractors and find that it uniformly improves their performance on the CUTE dataset in all categories except for the pose category. We hypothesize that this is due to the low difficulty of the pose set (as objects are rotated by only 15 degree increments). Because of this the non-cropped model achieves good performance on this subset as well, and the small imperfections of the cropping model aggregate to make the cropped performance slightly worse.
>
> > Size of the dataset
>
> As described in the global response, we have expanded the dataset during the rebuttal period to include 80 more objects of 10 categories (8 more instances in additional to the original 2 instances for each category, such that each category has 10 instances), as we agree that more instances for a small number of classes adds to the dataset’s value. The categories are `apple, banana, pen, mug_ceramic, fork, spoon, book, mouse, screwdriver, screw`. We show examples of the newly added data in the new Figure 4 of the revised manuscript. Note that this increases the number of total unique objects in the dataset from 100 to 180, and the total number of images in the dataset from 10,000 to 18,000, an 80% increase from the initial version at submission.
>
> > In-the-wild images and pose changes
>
> We agree that expanding the set of in-the-wild images and number of poses for each object would improve the dataset. This dataset is an initial effort towards providing a test set for evaluating the Platonic Distance metric, and curating and collecting data for each object class requires significant manual effort. We hope that future expansions can continue to grow this type of data in those directions.
>
> > Exploring other problems surrounding object similarity
>
> We agree that there are many aspects to object similarity. One can measure the visual similarity - such as the shape, color or texture of objects or functional similarity such as the purpose or affordance of an object. Often these measures are entirely orthogonal to each other, and further influenced by the context of the comparison. Because of this, many measurements of object similarity lack proper ground truth. Our aim is to define one particular dimension of similarity where we can obtain at least partial ground truth labels. We can obtain strong binary labels for this particular metric based on the identity of the objects themselves -- different images of the same object should have an ideal perfect similarity.

---

> ### Author Response · Authors · 2023-08-24
> **Response To Reviewer Ewhk (2/2)**
>
> > Additional vehicle re-identification comparison:
>
> We have updated the vehicle re-identification experiments to include the comparison of SpCL + LPIPS. We found that SpCL + FFA outperforms the combination of SpCL + LPIPS, and in fact, on the validation set we found that SpCL alone outperformed any weighted combination of SpCL and LPIPS we tested. We have also added an additional baseline that directly uses DINOv2 global features for ReID, and an ablation of DINOv2 to DINOv1 features. The results are as follows, and are reported also in Table 3 in the paper:
>
> | **Metric** | **VeRi** | | **CityFlow** | |
> |-----------------------|:--------:|:-------:|:-----------:|:-------:|
> | | top-1↑ | top-5↑ | top-1↑ | top-5↑ |
> | SpCL | 43.0 | 57.8 | 26.7 | 35.2 |
> | LPIPS | 22.5 | 36.1 | 3.5 | 8.0 |
> | DINOv2 (Global) | 16.0 | 26.5 | 15.3 | 23.0 |
> | SpCL+LPIPS | 41.6 | 56.5 | 24.0 | 33.1 |
> | SpCL+FFA DINOv1 (Crop-Img) | 43.6 | 58.1 | 23.9 | 32.7 |
> | SpCL+FFA (Crop-Img) (Ours) | 44.0 | 58.2 | 27.7 | 35.8 |
>
> > Fundamental limitations of the dataset and similarity metric
>
> Thank you for raising this point – aside from the limitations mentioned of lack of images depicting occluded objects and that poses of objects on the turntable are not varied, our dataset also contains only images of centered objects that take up a majority of the frame. Their capture conditions are also limited in terms of geographic and outdoor lighting conditions, and while we attempt to vary the capture devices themselves, only 4 total devices are used. We have updated the limitations section of the manuscript with this discussion of these fundamental limitations.
>
> > Crowdsourcing or research with human subjects
>
> We have corrected checklist item 5 to state “Yes”. We have also added the full text provided to participants in the updated supplementary document. IRB approval was not required as this is quality assessment rather than human research – our experiment was not interested in individuals nor did we collect any identifying information. We use the human input as a pseudo-ground truth for our comparison. We have updated checklist item 5 to include this information.
>
> > Clarity:
>
> Thank you for the helpful suggestions to improve the clarity of our paper. We have made edits accordingly:
>
> Tables 1 and 2: We have moved them.
> L212: We have amended the text to state that the top-1 accuracy is above 60% for the illumination setting and 90% for the pose setting for both methods.
> Section 4.1: We have merged it into the discussion in the introduction.
> L195-207: We have adjusted the text to use the phrase “we iterate over every image as the query” consistently.
> L235: We have summarized this paragraph into “For neural networks, we encode each image and measure which embedding lies farthest from the others and select it as the odd image out.”

---

> > ### Comment · Reviewer_Ewhk · 2023-08-25
> >
> > I thank the authors for providing a detailed response to concerns.
> >
> > The authors have addressed a number of my concerns, in particular for "Analysis of Foreground Masking," "Vehicle Re-identification," "Fundamental Limitations," "Crowdsourcing," and "Clarity." I believe the quantitative results from foreground masking and vehicle re-identification are compelling.
> >
> > Though I think that the size of the dataset is still somewhat small, I appreciate that the authors have been proactive with regards to expanding it. Given the recent trend of evolving datasets and the authors' continued data collection efforts after submission, it would be nice to see the authors commit to continue to extend it after publication, at least to the point where all classes are balanced. Though I understand that the dataset is meant to illustrate the PD, I maintain that the in-the-wild subset of the dataset is likely too small to be practically useful and thus doesn't serve a particular purpose.
> >
> > I do not believe that the authors have convincingly addressed my concerns of novelty in their response. I understand that the authors have tested many combinations of feature extractors and background cropping, which is an admirable effort. However, I am still unconvinced that taking an existing visual feature extractor (DINOv2) and combining it with an off-the-shelf foreground segmentation model constitutes a novel similarity metric. After some internal debate it's my opinion that the authors should lessen the claim of true novelty in favor of practicality, i.e., the 1/3 contribution (where the other 2/3s contribution is the dataset and the benchmark) is not so much a particular novel metric, but a survey of many admittedly simple metrics to find the one that works the best. If the authors are willing to somewhat soften their claim of a novel metric in the introduction, I would be willing to increase my score.

---

> > > ### Author Response · Authors · 2023-08-27
> > >
> > > Thank you for your prompt response and additional thoughtful feedback.
> > >
> > > > Dataset size
> > >
> > > We agree that a larger dataset will lead to improved practicality. We will make efforts to continue to expand the dataset in a systematic way.
> > >
> > > > Concerns of novelty:
> > >
> > > To address your concerns, we have significantly revised the text of the abstract and introduction to make it clear that our main contribution aside from the dataset and benchmark is a survey of many metrics in order to find a strong baseline for this task, which we denote as foreground feature averaging with DINOv2 features. This includes, but is not limited to, the following revisions:
> > >
> > > Introduction:
> > >
> > > Before:
> > > ```
> > > In this paper, we propose a surprisingly simple approach to approximately measuring Platonic distances. Our approach leverages the visual features of deep networks trained by contrastive self-supervised learning… With these simple designs, we demonstrate a powerful approach to approximately measuring the Platonic Distance.
> > > ```
> > >
> > > After:
> > > ```
> > > Using the CUTE dataset, we analyze existing similarity metrics and a simple but surprisingly effective framework for approximately measuring intrinsic object similarity…  We conduct an in-depth investigation into which pre-trained features and foreground extraction methods provide the best measure of this distance. From our results, we propose a strong baseline that incorporates two key ingredients that lead to a compact object-centric measurement…
> > > ```
> > >
> > > And in the paragraph summarizing the contributions:
> > >
> > > Before:
> > > ```
> > > We demonstrate that an approach combining foreground filtering and deep self-supervised features best measures PD among existing methods.
> > > ```
> > >
> > > After:
> > > ```
> > > We then perform an empirical study of metrics based on pre-trained visual feature extractors to determine which is most effective at measuring this similarity. From this, we propose a simple but strong baseline to measure the similarity called foreground feature averaging (FFA)...
> > > ```
> > >
> > > Abstract:
> > > Before:
> > > ```
> > > … we propose a simple yet effective approach to measuring PD based on deep features learned from contrastive self-supervised learning… We show that our approach best measures PD among current methods.
> > > ```
> > >
> > > After:
> > >
> > >
> > > ```
> > > … we find that combining deep features learned from contrastive self-supervised learning with foreground filtering is a simple yet effective approach to approximating the similarity. We conduct an extensive survey of pre-trained features and foreground extraction methods to arrive at a strong baseline that best measures intrinsic object-centric image similarity among current methods.
> > > ```
> > >
> > > Please don’t hesitate to leave any further comments regarding our response. We are more than happy to further discuss potential improvements.

---

> > > > ### Comment · Reviewer_Ewhk · 2023-08-27
> > > >
> > > > Thank you for the changes emphasizing the survey aspect of your contribution. Though I still have some concerns as to overall novelty I appreciate the authors' updates, and I understand that the findings will likely be helpful to researchers in the field of shape similarly. Therefore I have increased my score.

---

> > > > > ### Author Response · Authors · 2023-08-30
> > > > > **Thank you for your feedback!**
> > > > >
> > > > > We would like to thank you again for your thoughtful feedback and suggestions that have helped us improve our paper.  We are glad to see that you find the response helpful and the findings useful to researchers in the related field.  As the discussion period is coming to a close, we would like to ask if you may have any additional remaining concerns?  If so, we would be happy to discuss more and address them.

---

### Author Response · Authors · 2023-08-24
**Global Response**

Thank you to all of the reviewers for your detailed and thoughtful feedback. It has helped us to significantly improve the paper and dataset. We respond to each of the individual points and questions in each response, but include updates that address multiple reviewers’ concerns here. We have also uploaded a revised version of the manuscript, with changes in blue for convenience.

# Expanded Dataset
Based on feedback from the reviewers, we have expanded the dataset during the rebuttal period to include 80 more objects of 10 categories (8 more instances in additional to the original 2 instances for each category, such that each category has 10 instances), as we agree that having more instances from particular categories provides valuable coverage of intrinsic intra-class variation. The categories are `apple, banana, pen, mug_ceramic, fork, spoon, book, mouse, screwdriver, screw`. We show examples of the newly added data in the new Figure 4 in the updated manuscript. Note that this increases the number of total unique objects in the dataset from 100 to 180, and the total number of images in the dataset from 10,000 to 18,000, an 80% increase from the version at submission.

# Further Benchmarking of PD on the CUTE Dataset
Our goal with this work is to present a metric of similarity (i.e., the Platonic Distance), to benchmark different methods (including a strong baseline that we propose) to compute the metric, and to collect a dataset for such benchmarking. After reading the reviewers’ feedback, we performed an in-depth investigation into which features and foreground extraction method provides us the best measure of platonic distance. To disambiguate between the conceptual Platonic Distance and the model we describe in the paper, we henceforth use the name Foreground Feature Averaging (FFA) to describe the model that aggregates DINOv2 foreground features, as suggested by reviewer isG4.

We evaluated the following models as feature extractors: DINOv1, DINOv2, CLIP, and Unicom (a state-of-the-art image retrieval model) [1]. For each of these models we tested the performance with and without background cropping. We also tested different methods of integrating background cropping into our DINOv2 pipeline: applying the crop mask to the image directly before feature extraction (Crop-Img) or applying the mask to features after extraction (Crop-Feat). Finally, we evaluated a method from concurrent work, DreamSim [2], on our dataset.

The results are shown in the revised table 1 of the manuscript and below. We determine that DINOv2 features are the strongest out of the considered models. We find that utilizing the class token yields the strong performance on the varying pose subset, while cropping and taking the class token yields best in-the-wild results. The foreground feature averaging method (Crop-Feat) still performs best overall, ranking first in 5 of the 12 evaluations and performing competitively in others.

| **Setting**  | **In-the-wild** | |  | **Illumination** | |  | **Pose** | |  | **All** | |  |
|---------------------------|:---------------:|:------:|:----------:|:----------------:|:------:|:----------:|:-------:|:------:|:----------:|:-------:|:------:|:----------:|
|    | mAP↑  | top-1↑ | ARI↑ | mAP↑  | top-1↑ | ARI↑ | mAP↑ | top-1↑ | ARI↑ | mAP↑ | top-1↑ | ARI↑ |
| LPIPS   | 38.7 | 49.1 | - | 41.6 | 53.9 | - | 38.2 | 49.3 | - | 39.6 | 49.8 | - |
| SSIM   | 40.1 | 49.5 | - | 39.4 | 52.1 | - | 40.3 | 50.6 | - | 39.9 | 50.0 | - |
| CLIPScore   | 54.4 | 13.3 | -8.7 | 62.0 | 38.8 | -1.0 | 82.8 | **98.2**| 48.6 | 65.2 | 94.9 | -0.1 |
| CLIPScore + Crop  | 69.3 | 59.5 | 15.4 | 69.3 | 75.0 | 11.5 | 77.7 | 94.7 | 38.7 | 68.1 | 94.5 | 18.6 |
| DreamSim   | 56.9 | 11.8 | -6.3 | 60.7 | 38.5 | -11.0 | 82.7 | 96.5 | 48.2 | 64.3 | 93.2 | 3.6 |
| DreamSim + Crop  | 66.8 | 47.0 | 6.0 | 69.4 | 72.0 | -4.0 | 80.6 | 96.4 | 39.9 | 66.8 | 95.0 | 3.7 |
| DinoV2   | 70.9 | 47.3 | 19.8 | 86.0 | 82.3 | 54.0 | **85.2**| 97.2 | **56.1** | **78.4**| 95.7 | 40.3 |
| DinoV2 + Crop  | **79.1** | **69.0** | **41.0** | 84.0 | 85.5 | 39.8 | 83.1 | 96.4 | 53.0 | 76.2 | 96.1 | 36.6 |
| DinoV1   | 52.6 | 3.8 | -8.9 | 58.6 | 37.0 | -10.8 | 81.8 | 97.2 | 47.2 | 63.6 | 93.5 | 1.8 |
| DinoV1 + Crop  | 64.4 | 37.8 | 7.1 | 67.0 | 63.0 | -4.7 | 79.6 | 96.5 | 38.0 | 66.4 | 94.8 | 4.0 |
| Unicom   | 63.4 | 37.0 | 4.8 | 64.9 | 46.5 | 6.7 | 81.9 | 97.1 | 48.5 | 66.6 | 94.8 | 9.7 |
| Unicom + Crop  | 73.9 | 66.3 | 25.5 | 72.2 | 72.3 | 11.3 | 78.0 | 93.8 | 39.5 | 67.9 | 93.5 | 15.1 |
| FFA Crop-Img (ours) | 76.7 | 68.3 | 35.5 | 81.7 | 83.0 | 48.2 | 79.1 | 95.2 | 41.5 | 72.2 | 94.9 | 26.8 |
| FFA Crop-Feat (ours) | 75.1 | 61.8 | 27.2 | **88.8** | **89.0**| **62.9** | 81.9 | 96.6 | 46.8 | 77.1 | **96.2**| **40.5** |


[1]: Unicom: Universal and Compact Representation Learning for Image Retrieval. An et al, ICLR 2023

[2]: DreamSim: Learning New Dimensions of Human Visual Similarity using Synthetic Data. Fu et al., 2023

---

### Decision · Program_Chairs · 2023-09-22

**Decision:**

Accept (Poster)

**Comment:**

Three reviewers support the acceptance of this paper. One reviewer Ewhk remains marginally below the acceptance threshold at 5 after improving the score from the original score. The authors have provided extensive responses and discussions to address the issues raised. The remaining two issues Ewhk highlighted are the small size of the dataset especially from in-the-wild, and weak justification on the novelty of the proposed similarity metric. Overall, all four reviewers acknowledged that the authors' extensive responses and clarifications have improved the paper notably. I also share the view that the similarity metric per se is not central to the contribution of this paper and appreciate that the authors have moderated their claims on this novelty. I also share the view that the dataset is indeed small as it stands, and in particularly on in-the-wild setting, in order for it to be more widely useful in practice. Having said that, the authors are aware of this weakness and are making a proactive effort to continuously increase the scale of the dataset. On balance, I recommend accept.